# Flexynesis: A deep learning toolkit for bulk multi-omics data integration for precision oncology and beyond

Bora Uyar [1] ✉, Taras Savchyn[1], Amirhossein Naghsh Nilchi [2,3], Ahmet Sarigun[1], Ricardo Wurmus [1], Mohammed Maqsood Shaik[1], Björn Grüning [2], Vedran Franke [1] & Altuna Akalin [1] ✉

Accurate decision making in precision oncology depends on integration of multimodal molecular information, for which various deep learning methods have been developed. However, most deep learning-based bulk multi-omics integration methods lack transparency, modularity, deployability, and are limited to narrow tasks. To address these limitations, we introduce Flexynesis, which streamlines data processing, feature selection, hyperparameter tuning, and marker discovery. Users can choose from deep learning architectures or classical supervised machine learning methods with a standardized input interface for single/multi-task training and evaluation for regression, classification, and survival modeling. We showcase the tool's capability across diverse use-cases in precision oncology. To maximize accessibility, Flexynesis is available on PyPi, Guix, Bioconda, and the Galaxy Server (https://usegalaxy.eu/). This toolset makes deep-learning based bulk multi-omics data integration in clinical/pre-clinical research more accessible to users with or without deep-learning experience. Flexynesis is available at https://github.com/BIMSBbioinfo/flexynesis.

Cancer is a complex disease primarily resulting from genomic aberrations. The disease is marked by abnormal cell growth, invasive proliferation, and tissue malfunction, impacting twenty million individuals and causing ten million yearly deaths worldwide[1]. To bypass protective mechanisms, cancer cells must acquire several key characteristics, such as resistance to cell death, immune evasion, tissue invasion, growth suppressor evasion, and sustained proliferative signaling[2]. Unlike rare genetic disorders, caused by few genetic variations, complex diseases, like cancer, require a comprehensive understanding of interactions between various cellular regulatory layers. This entails data integration from various omics layers, such as the transcriptome, epigenome, proteome, genome, metabolome, and microbiome[3]. In clinical settings, genome-informed diagnostics to identify disease-causing variants are already in use[4]. However, capturing the complexity of most cancers requires more than a panel of genomic markers. Multi-omics profiling is a vital step toward understanding not only cancer but other complex diseases like cardiovascular and neurological disorders[5–7]. Proof-of-concept studies have shown the benefits of multi-omics patient profiling for health monitoring, treatment decisions, and knowledge discovery[8]. Recent longitudinal clinical studies in cancer are evaluating the effects of multi-omics-informed clinical decisions compared to standard of care[9]. Addressing this need for multi-omic profiling to improve the understanding of complex diseases, major international initiatives have developed multi-omic databases such as The Cancer Genome Atlas (TCGA), the Cancer Cell Line Encyclopedia (CCLE)[10] to enhance molecular profiling of tumors and disease models.

[1]Bioinformatics and Omics Data Science Platform, Max Delbrück Center for Molecular Medicine, The Berlin Institute for Molecular Systems Biology, Hannoversche Str. 28, 10115 Berlin, Germany. [2]Bioinformatics Group, Department of Computer Science, Albert-Ludwigs-University Freiburg, Georges-Köhler-Allee 106, 79110 Freiburg, Germany. [3]Institute of Experimental and Clinical Pharmacology and Toxicology, Faculty of Medicine, University of Freiburg, Freiburg, Germany. ✉e-mail: bora.uyar@mdc-berlin.de; Altuna.Akalin@mdc-berlin.de

While cell regulation at the molecular level is highly interconnected, redundant, and has non-linear relationships between components, the information about these intricate relationships is usually isolated in different molecular data modalities. Each molecular profile is measured one assay at a time (as in assays developed for profiling the transcriptome, the genome, the methylome etc), however, all the different layers of molecular information are in actuality in a cross-talk with one another. Therefore, it is important to capture the non-linear relationships, and impacts of disruptions of the different components of the cellular machinery by combining the disparate data modalities into a more meaningful synthesis. However, the high dimensionality of molecular assays and heterogeneity of the studied diseases create computational challenges.

The challenges of multi-omics data integration prompted development of various machine learning algorithms, including deep learning approaches[11,12]. Available benchmarking studies that compared different deep-learning-based methods for multi-omics integration for classification and regression tasks[13,14] have shown that none of the methods clearly outperformed others in all the tasks at hand. This necessitates a flexible and reproducible approach that provides adaptable architectures for solving each computational task.

Before setting out to develop yet another deep learning-based multi-omics integration method, despite the availability of the myriad of published studies[11], we collated a survey of available bulk multi-omics data integration methods to see which tools can be easily adapted for our own translational research projects (Supplementary Data 1). Such projects usually include heterogeneous cohorts of cancer patients and pre-clinical disease models with multi-omics profiles. A primary issue we observed with existing methods is their limited reusability or adaptability to different datasets and contexts. The majority of published approaches do not provide accompanying code, severely limiting their accessibility and applicability. Even when the code is available, it often exists as an unpackaged collection of scripts or notebooks. Such a disorganized format makes these methods difficult, if not impossible, to install, reuse, and incorporate into existing bioinformatics pipelines. Out of the 80 studies collated, 29 studies provide no codebase. 45 studies provide collections of scripts/notebooks, with the goal of reproducing the findings in the published study rather than serving as a generic tool for multi-omics integration. While these methods (Supplementary Data 1) are valuable contributions to the scientific community, they still require extensive customization to make them usable for different datasets and tasks.

Besides lacking readily available code, published methods suffer from one or more of the criteria that are crucial for ensuring the reliability and reproducibility of machine learning applications. Standard operating procedures such as training/validation/test splits, hyperparameter optimization, feature selection, and marker discovery are frequently overlooked or manually defined, without any accompanying documentation again underscoring the arduous amount of work needed to adapt these approaches for custom problems.

Another limitation of current deep learning methods is their narrow task specificity. Many tools are designed exclusively for specific applications, such as regression, survival modeling, or classification. Comprehensive multi-omics data analysis frequently requires a mixture of such tasks, however, the specialization of already existing tools restricts their applicability.

While deep learning methods are sometimes considered as superior, classical machine learning algorithms frequently outperform them[15–17]. This performance differential is not immediately apparent, and often not tested with the currently existing tools, requiring users to undertake extensive benchmarking to uncover the most effective solution to their specific problem.

Addressing these challenges, we introduce Flexynesis, a deep learning framework for multi-omics data integration designed to overcome the above-mentioned limitations (Fig. 1). We demonstrate the versatility of Flexynesis through various use cases, including drug response prediction, cancer subtype modeling, survival analysis, and biomarker discovery. We demonstrate how to handle multiple tasks simultaneously, supporting a combination of regression, classification, and survival tasks. We show use-cases where the flexibility of neural networks can be utilized in different prediction tasks by building models of both unsupervised and supervised tasks, with one or more supervision heads, and symmetric (auto-encoders) and asymmetric (cross-modality) encoder-decoder combinations. To further enhance its utility, we provide an accessory pipeline and a collection of datasets for benchmarking different flavors of Flexynesis. This benchmarking includes a comparison to classical machine learning methods (Random Forest, Support Vector Machines, XGBoost, and Random Survival Forest). In summary, the landscape of published deep learning methods for bulk multi-omics data integration is fraught with challenges that hinder their effective reuse and integration into broader bioinformatics workflows. This manuscript addresses these challenges and introduces Flexynesis, a comprehensive solution designed to enhance the utility and applicability of deep learning in multi-omics data analysis.

## Results

We designed Flexynesis for automated construction of predictive models of one or more outcome variables. For each outcome variable, a supervisor multi-layer-perceptron (MLP) is attached onto the encoder networks (a selection of fully connected or graph-convolutional encoders) to perform the modeling task. Clinically relevant machine learning tasks such as drug response prediction (regression), disease subtype prediction (classification), and survival modeling (right-censored regression) tasks are all possible as individual variables or as a mixture of variables, such that each outcome variable has an impact on the low-dimensional sample embeddings (latent variables) derived from the encoding networks (See Supplementary Figs. 1–8 for the schematic representation of different model architectures, workflows for data processing, hyperparameter optimisation, and model fine-tuning).

### Single-task modeling: predicting only one outcome variable

In Fig. 2, we demonstrate the different kinds of modeling tasks that are possible with Flexynesis using a single outcome variable (single MLP) as regression (Fig. 2A), classification (Fig. 2B), and survival models (Fig. 2C). For the regression task, we trained Flexynesis on multi-omics (gene expression and copy-number-variation) data from cell lines from the CCLE database[10] to predict the cell line sensitivity levels to the drugs Lapatinib, a tyrosine kinase inhibitor, and Selumetinib, a MEK inhibitor. We evaluated the performance of the trained model on the cell lines from the GDSC2 database[18] which were also treated with the same drugs, where we observed a high correlation between the known drug response values and the predicted response values for both drugs (Fig. 2A).

For the single-variable classification task, we demonstrate classification of seven TCGA datasets including pan-gastrointestinal and gynecological cancers with respect to their microsatellite instability (MSI) status using gene expression and promoter methylation profiles. MSI is a molecular phenotype that displays a high mutational load that results from deficient DNA mismatch repair mechanisms[19]. Moreover, high-MSI levels are predictive of response to immune checkpoint blockade therapies[20], underscoring the relevance of detecting MSI-High samples. As MSI-High is characterized by a high mutational load, it would not be surprising to achieve a good classification performance to predict the MSI status using mutation data. We demonstrate that, without using the mutation data, we can achieve a very high accuracy

classifier (AUC = 0.981) using gene expression and methylation profiles (Fig. 2B). We have also benchmarked multiple deep learning architectures and data type combinations and observed that the best performing model was trained on gene expression data only (Supplementary Data 2). This result suggests that samples that have been profiled using RNA-seq, but lack genomic sequencing data could still be classified in terms of MSI status.

As the third type of modeling task, we demonstrate survival modeling using Flexynesis on a combined cohort of lower grade glioma (LGG) and glioblastoma multiforme (GBM) patient samples[21]. For survival modeling, a supervisor MLP with Cox Proportional Hazards loss function is used to guide the network to learn patient-specific risk scores based on the input overall survival endpoints as has been demonstrated previously[22]. After training the model on 70% of the samples, we predicted the risk scores of the remaining test samples (30%) and split the risk scores by the median risk value in the cohort. The embeddings visualized based on the median risk score stratification shows that the test samples are clearly separable in the sample embedding space, which is also confirmed by the Kaplan-Meier survival plot, which shows a significant separation of patients in terms of predicted risk scores (Fig. 2C).

## Multi-task Modeling: Joint prediction of multiple outcome variables

While being able to build deep learning models with any of the regression/classification/survival tasks individually offers an improved user experience, this is also usually possible with classical machine learning methods. The actual flexibility of deep learning is more evident in a multi-task setting where more than one MLPs are attached on top of the sample encoding networks, thus the embedding space can be shaped by multiple clinically relevant variables. This flexibility is even more pronounced in the presence of missing labels for one or more of the variables, which is tolerated by Flexynesis.

To demonstrate the use of multi-task modeling, we trained models on 70% of the METABRIC dataset (a metastatic breast cancer cohort with multi-omics profiles of 1865 patients)[23] and obtained the embeddings for the 30% of the samples. In order to compare and contrast the effect of multi-task modeling with single-task modeling, we chose two clinically relevant variables for this cohort: subtype labels (CLAUDIN_SUBTYPE) and chemotherapy treatment status (CHEMOTHERAPY). We built three different models: a single-task model using only the subtype labels (Fig. 3A), a single-task model using

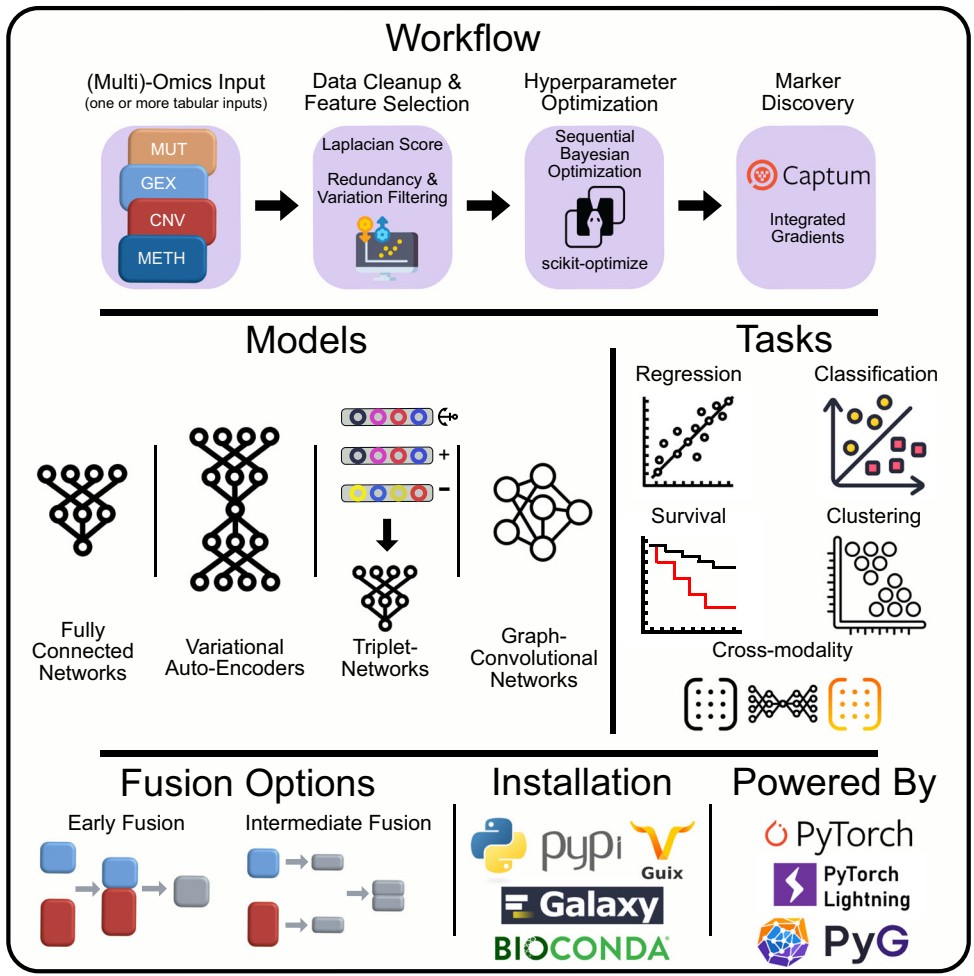

**Fig. 1 | Summary of the Flexynesis data integration and analysis workflow.** Flexynesis accepts as input one or more data tables in tabular format along with sample metadata, carries out various data-cleaning steps, and provides multiple feature-selection options. A sequential Bayesian hyperparameter optimization routine is applied for model training. The trained models are evaluated on the test data using a variety of metrics, and input features are assessed and ranked based on feature attribution scores with respect to their contribution to outcome prediction tasks. Flexynesis currently supports multiple neural network architectures, including classical feedforward neural networks, variational autoencoders, multi-triplet neural networks, and graph-convolutional neural networks. Each network can be utilized in a supervised multi-task setting for regression, classification, or survival analysis, as well as in unsupervised or cross-modality prediction tasks. Separate data modalities can be fused using either early or intermediate fusion options. Flexynesis is available for installation through publicly accessible repositories such as PyPI, Guix, and Bioconda, and is ready to be used on the Galaxy platform. Flexynesis is developed using PyTorch, PyTorch Lightning, and PyTorch Geometric.

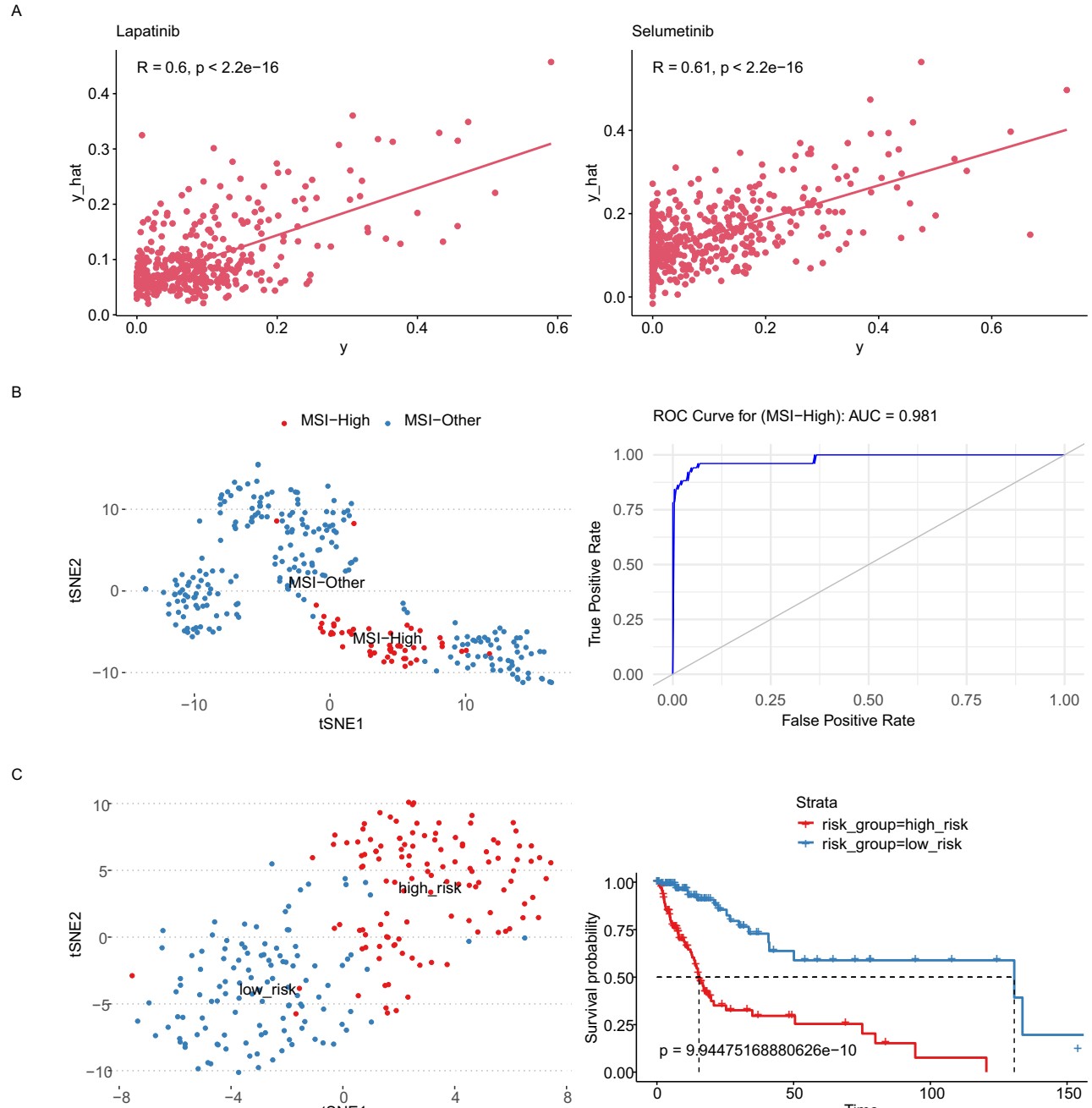

**Fig. 2 | Flexynesis supports single-task modeling for regression (A), classification (B), and survival (C).** For all three tasks, both a fully-connected-network and a supervised variational auto-encoder was trained and best-performing model's results were presented. **A** Performance evaluation of Flexynesis on drug response prediction of a model trained on 1051 cell lines from CCLE (using RNA and CNV profiles) and evaluated on 1075 cell lines from GDSC2 for the drugs Lapatinib (Pearson correlation test, $r = 0.6$, $p = 7.750175e\text{-}42$) and Selumetinib (Pearson correlation test, $r = 0.61$, $p = 3.873949e\text{-}50$). The x-axis depicts observed drug response values (AAC-recomputed as in Pharmacogx package)[60] and the y-axis depicts the predicted drug response values for the test samples. **B** Evaluation of Flexynesis on microsatellite instability (MSI) status prediction using gene expression and/or promoter methylation data from seven different TCGA cohorts (gastrointestinal and gynocological cancers) with microsatellite instability (MSI) annotations: TCGA-COAD (Colon Adenocarcinoma), TCGA-ESCA (Esophageal Carcinoma), TCGA-PAAD (Pancreatic Adenocarcinoma), TCGA-READ (Rectum Adenocarcinoma), TCGA-STAD (Stomach Adenocarcinoma), TCGA-UCEC (Uterine Corpus

Endometrial Carcinoma), TCGA-UCS (Uterine Carcinosarcoma). The models were trained on 70% of the samples ($N = 1133$) with MSI status annotations and evaluated on the remaining 30% of the samples ($N = 283$). The tSNE (t-distributed Stochastic Neighbor Embedding) plot represents the sample embeddings colored by MSI status and the ROC curve represents the best performing deep learning model based on both gene expression and methylation data. **C** Evaluation of Flexynesis on a survival modeling task on a merged cohort of LGG (Lower Grade Glioma) and GBM (Glioblastoma Multiforme) (using mutations and copy-number-alteration profiles). The model is trained on 557 samples and evaluated on 239 test samples. The tSNE plot depicts the sample embeddings obtained from the model encoder for the test samples colored by the predicted Cox proportional hazard risk scores stratified into "high-risk" and "low-risk" based on the median risk score. The Kaplan-Meier-Plot represents the survival stratification of the test samples based on this risk stratification (Logrank Test, $p = 9.94475168880626e - 10$). Source data are provided as a Source Data file.

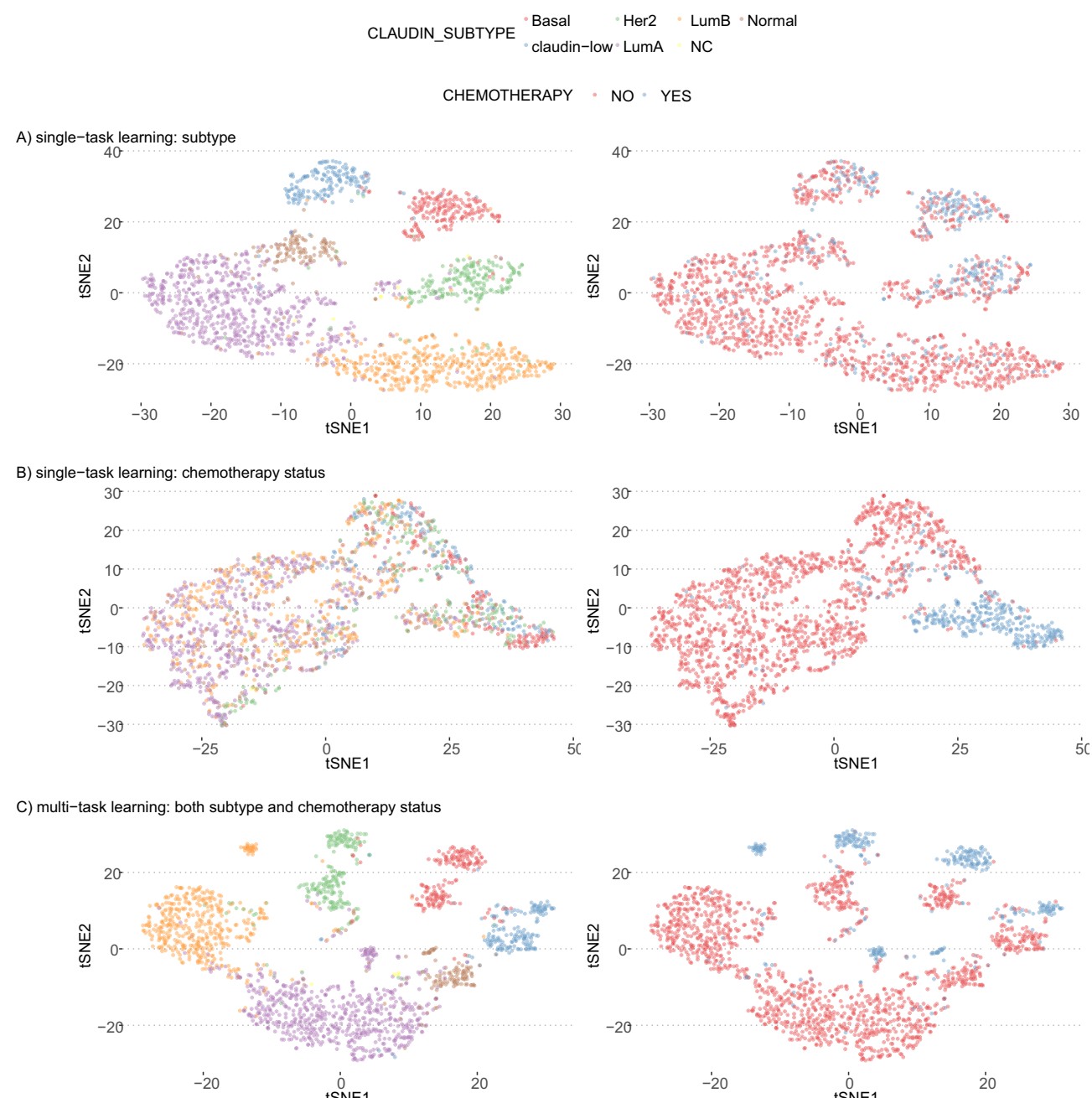

**Fig. 3 | t-SNE plots representing sample embeddings of 1865 metastatic breast cancer samples from the METABRIC study.** These plots compare the impact of single-task and multi-task modeling on the clustering of samples by clinical variables. Plots on the left are colored by the breast cancer subtype and the plots on the right are colored by the treatment status. **A** Single-Task Model – Breast Cancer Subtypes: *t*-SNE (*t*-distributed Stochastic Neighbor Embedding) visualization of sample embeddings obtained from a single-task model trained exclusively to predict breast cancer subtypes. **B** Single-Task Model – Chemotherapy Status: *t*-SNE plot visualization of sample embeddings from a model trained only to predict the chemotherapy status of patients, showing the segregation capability of the single-task model with respect to treatment status. **C** Multi-Task Model – Subtypes and Chemotherapy Status: *t*-SNE plot of sample embeddings from a multi-task model trained with dual supervisor heads: one for breast cancer subtypes and another for chemotherapy status. The plot shows how multi-task learning influences the embedding space, enhancing the separation of samples based on both clinical variables simultaneously. Source data are provided as a Source Data file.

only the chemotherapy status of the patients (Fig. 3B), and finally a multi-task model using both subtype labels and chemotherapy status as outcome variables (Fig. 3C). Coloring the samples by the subtype labels and chemotherapy status, we can observe that the sample embeddings obtained exclusively for the subtype modeling reflect a clear clustering of samples by subtype, but not by the chemotherapy status (Fig. 3A). Similarly, the sample embeddings obtained from the model trained exclusively with the chemotherapy status as outcome variable shows a clear separation of samples by treatment status, however the separation by subtypes is not as evident anymore (Fig. 3B). In the multi-task setting where the model had two MLPs (one for subtype labels and one for chemotherapy status), the sample embeddings show a clear separation of both by the subtype labels and also the chemotherapy status (Fig. 3C).

We also analyzed the LGG and GBM cohort (from Fig. 2C) in a multi-task setting where we attached three separate MLPs on the

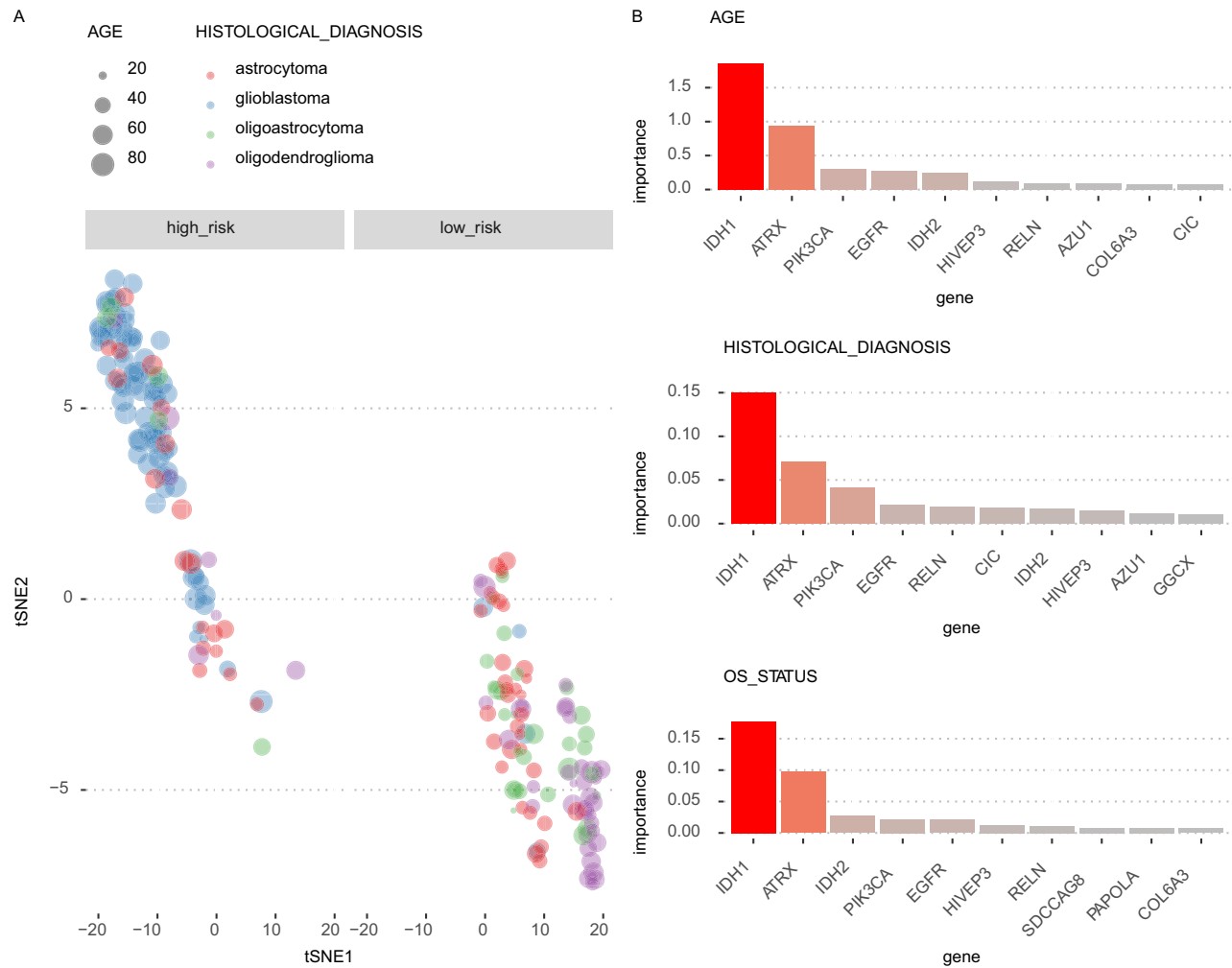

**Fig. 4 | Flexynesis can be trained concurrently for all three types of tasks: regression, classification, and survival at a single run.** The model was trained on 557 training samples from the merged cohort of the LGG (Lower Grade Glioma) and GBM (Glioblastoma Multiforme) patient samples with three supervisor heads: a regressor for the patient age (AGE), a classifier for the histological diagnosis, and a survival head for the overall survival status of the patient (OS_STATUS). **A** Displays the tSNE (*t*-distributed Stochastic Neighbor Embedding) visualization of the sample embeddings for 239 test samples, where the size of the points reflect the age of the patient, the colors represent the histological diagnosis, and the samples were stratified into high-risk and low-risk groups based on the predicted risk scores for each patient. The sample embeddings reflect the impact of all three clinical variables concurrently. **B** Displays the top 10 most important features discovered for each supervisor head for the patient's age, histological subtype, and survival status. Source data are provided as a Source Data file.

encoder layers: a regressor to predict the patient's age (AGE), a classifier to predict the histological subtype (HISTOLOGICAL DIAGNOSIS), and another survival head to model the survival outcomes of the patients (OS_STATUS). Concurrently training the model with three different tasks at the same time, we inspected the sample embeddings and observed that older patients with high risk scores have the glioblastoma subtype, while younger patients with lower risk scores have the other subtypes, where low risk young patients can still be distinguished mainly by histological subtype (Fig. 4A). Thus, training the model on three clinically relevant variables helps us obtain sample embeddings that reflect all three variables in a hierarchical manner. Inspecting the top markers for each of these variables, we observe common genes for all three variables such as *IDH1*, *IDH2*, *ATRX*, *PIK3CA*, and *EGFR* (Fig. 4B), which could be explained by the fact that the clinical variables such as age and histological subtype are correlated with the survival outcomes of the patients, underpinning the importance of these genes in the etiology of the gliomas, which have been extensively studied and reported before[24].

**Unsupervised learning: finding groups and general patterns**

One of the main architectures provided in Flexynesis is the variational auto-encoders (VAE) with maximum mean discrepancy (MMD) loss[25]. While VAEs are usually employed in unsupervised training tasks, in Flexynesis they can be used for both supervised and unsupervised tasks. In the absence of any target outcome variables (in other words, without any additional MLP modules attached on top of the encoders), the network behaves as a VAE-MMD where the sole goal is to reconstruct the input data matrices, while generating embeddings that follow a Gaussian distribution due to the MMD loss.

As a proof of principle experiment, we trained a VAE-MMD model without any attached supervisor MLPs, to test the unsupervised dimension reduction capabilities on 21 cancer types from the TCGA resource using gene expression and methylation as input modalities. Applying k-means clustering (k from 18 to 24), we obtained a clustering of the samples based on the trained sample embeddings. The tSNE representation of the resulting sample embeddings shows a clear separation of unsupervised clusters (Fig. 5A) and the known sample labels (Fig. 5B) with a good correspondence between unsupervised

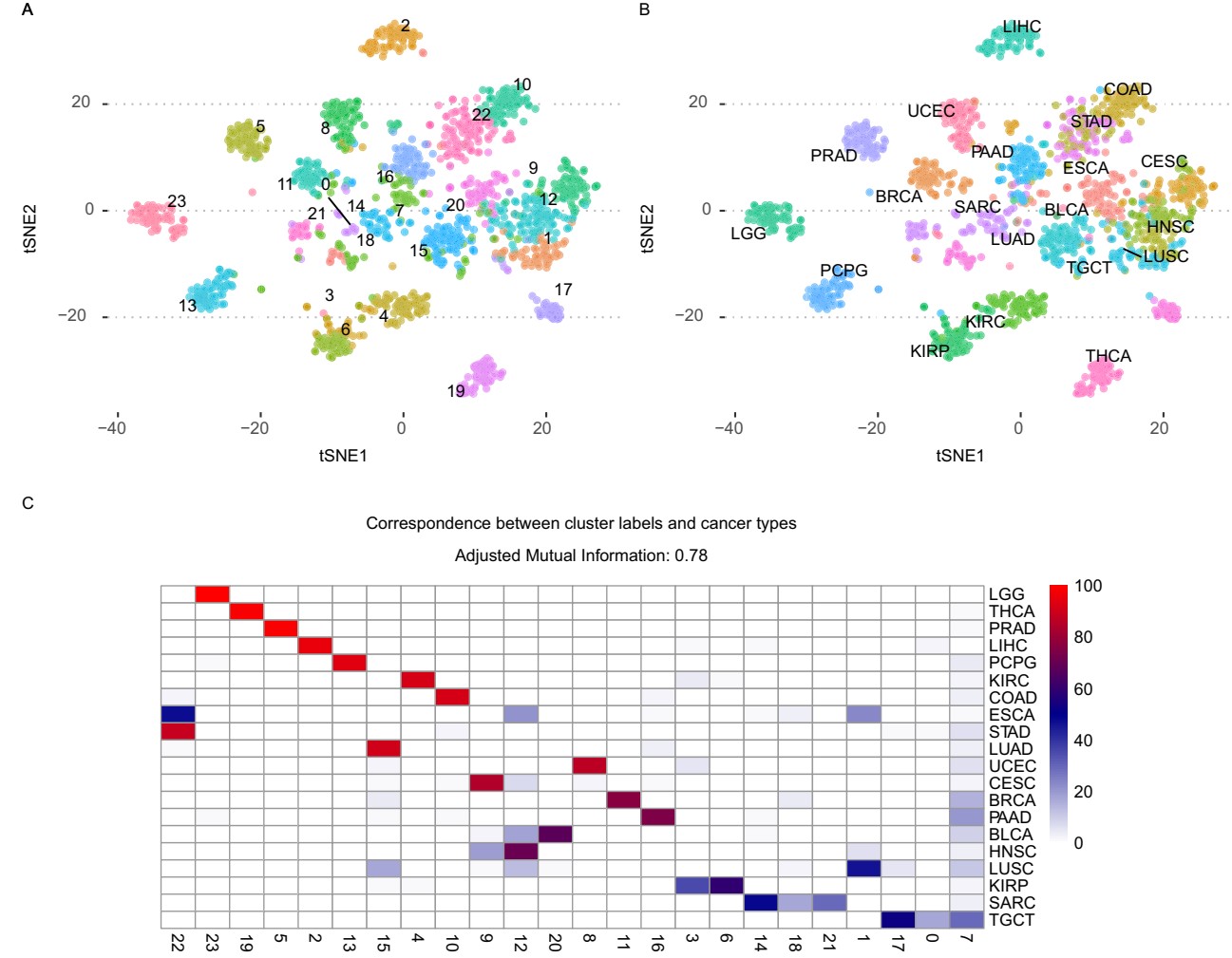

**Fig. 5 | Flexynesis can be used for unsupervised training and clustering.** The figure displays the unsupervised analysis of 21 cancer types from the TCGA study for 1600 samples (80 samples per cancer type were randomly selected). **A** Displays the tSNE plot of the training sample embeddings colored by the best performing clustering scheme using the *k*-means algorithm for values of $18 <= k <= 24$, where best clustering was selected by the best silhouette score ($k = 24$). **B** The same tSNE (*t*-distributed Stochastic Neighbor Embedding) plot as in (**A**) but colored by the known cancer type labels. **C** The heatmap displays the concordance between the cluster labels from (**A**) and known cancer type labels from (**B**), where the adjusted mutual information score is 0.78. Each row is normalized to add up to 100% where the color of the cells represent the concordance percentage of the cancer types to the corresponding cluster labels. Source data are provided as a Source Data file.

clusters and known sample labels (adjusted mutual information: 0.78) (Fig. 5C, Supplementary Data 3).

## Cross-modality learning: transferring knowledge between different omic data types

While variational autoencoders are designed to reconstruct the initial input data, this can be formulated in a different fashion such that the goal of the reconstruction is a set of matrices different from the inputs. Thus, it is possible to build models where the input data modalities differ from the output data modalities. For instance, a gene expression data matrix could be used to reconstruct a mutation data matrix, thus learning how to translate between these modalities, while simultaneously learning the low-dimensional embeddings that reflect this translation. Due to the modular structure of the Flexynesis, we can also attach one or more MLPs on top of these cross-modality encoder models for one or more target variables as supervisors for regression, classification, and survival tasks.

In order to demonstrate this feature, we designed an experiment using the genome-wide gene essentiality scores measured for >1000 cell lines as part of the DepMap project[26]. The DepMap database contains measurements of cellular proliferation after perturbation of all protein coding genes. It has been previously shown that for a given

cell line, the gene expression profiles of the cell lines can be used to predict the gene essentiality scores[27,28]. Here, we carried out a similar approach, where gene expression profiles of genes across cell lines were used as input with a goal to reconstruct the cancer cell line dependency scores of the same genes. We expanded this approach to a multi-modal setup, where we used two additional data modalities besides the gene expression: (1) we used pre-trained large language models to generate protein sequence embeddings for the same genes using Prot-Trans[29] and obtained sequence embedding vectors for each gene (using the canonical protein sequences) (2) we used the structural and functional features of proteins (such as disorder profiles, evolutionary sequence conservation, secondary structures, post-translational modification sites) from the DescribePROT database[30]. Thus, each gene was represented by three data modalities: gene expression profiles across cell lines, protein sequence embeddings, and describeProt features. We used these modalities to reconstruct the gene-essentiality scores for each of the cell lines in the DepMap database. In addition, we attached a supervisor MLP to guide the network to predict the hubness-score of each gene in the genetic interaction networks obtained from the STRING database[31] assuming that the centrality of a gene in biological interaction networks could be a contributing factor in its essentiality for cell survival (Fig. 6A). Thus, the

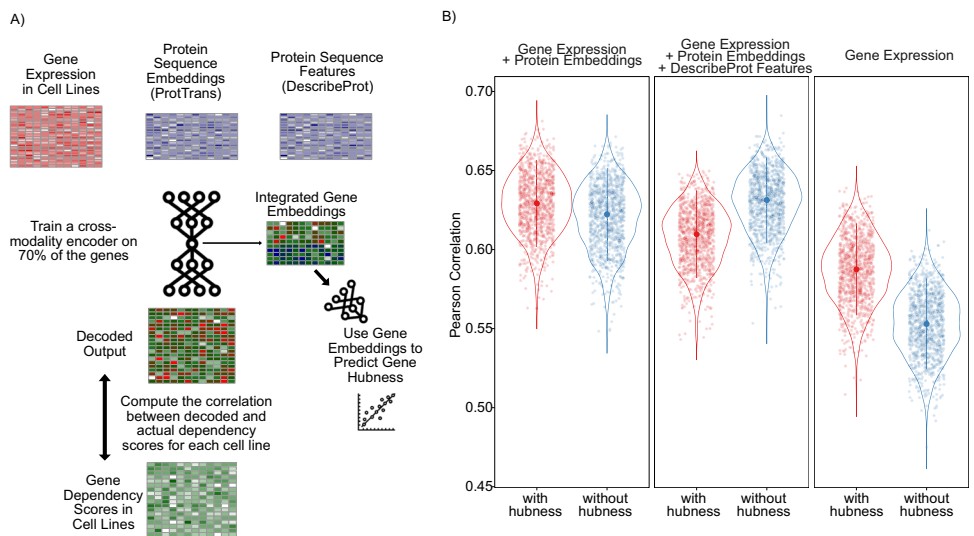

**Fig. 6 | Flexynesis can be used to build matrix-to-matrix (cross-modality) prediction models. A** Multi-modal cross-modality prediction of gene knock-out dependency probabilities of cell lines. A cross-modal encoder-decoder model was used that takes as input a combination of input data modalities and reconstructs the CRISPR-based gene knock-out dependency probability scores of cell lines from the DepMap project. Each gene is represented with a combination of feature sets including the expression profile of the gene across cancer cell lines, Prot-Trans large language model embeddings of the gene's canonical protein sequence, and functional/structural sequence features of the gene's canonical protein sequence from the describePROT database. The cross-modality encoder was trained both with and without an attached supervisor that predicts the hubness score of the gene according to its centrality in the STRING database. **B** The distribution of the correlation scores for each ($N=1064$) cell line's measured gene knock-out

dependency scores and the predicted scores (for the test set) based on different input data modality combinations: "Gene Expression" represents the prediction performance when using only gene expression profiles of the genes in the cell lines; "Gene Expression + Protein embeddings" represents the prediction performance when using both the gene expression profiles and protein sequence embeddings from Prot-Trans; "Gene Expression + Protein embeddings + DescribeProt Features" represents the prediction performance when using all three feature sets including gene expression, protein embeddings, and describeProt features. The center line in each violin represents the median, and the shaded regions span the interquartile range (IQR). Individual data points each representing one cell line ($N=1064$) are overlaid using jittered dots, color-coded by hubness category. Source data are provided as a Source Data file.

model was trained concurrently to predict both the gene essentiality score in a particular cell line (as a matrix), along with the gene hubness (as a vector). We trained the model on 70% of the genes and evaluated the model on the remaining 30% of the genes, by computing the average correlation of each cell line's predicted gene dependency scores with the measured scores. The addition of the protein sequence embeddings from the language models had a significant improvement on the performance of the model, while the addition of DescribeProt features did not make an additional improvement over the protein language embeddings (Fig. 6B), which suggests that LLM-based protein embeddings might be already capturing similar information to the features from describePROT. For comparison, we also built models without the supervision for the "hubness" feature. We observed that depending on the data combinations used as input, using a supervisor for "hubness" led to an improvement for the single data modality case (Fig. 6B, right panel) but also a deterioration for the reconstruction scores when all three modalities were used as input (Fig. 6B, middle panel), which could be because the network may have put more weight on learning the "hubness" feature while the weights on cross-modality reconstruction may have been diluted with the addition of further data modalities in this particular case.

### Improving model performance via model fine-tuning
One of the conveniences offered by neural networks compared to classical machine learning approaches is that the neural networks trained on a source dataset can be fine-tuned on a small portion of the target dataset. This feature offers a possibility to tune the trained model on the potentially shifted distribution of the target dataset compared to the source[32]. We implemented an optional fine-tuning procedure, which uses a portion of the test dataset to modify the model parameters (following a combination of model parameter

freezing strategies and different learning rates). The fine-tuned model is then evaluated on the remaining test dataset samples. In the first experiment, we trained multiple neural network models along with baseline methods (Random Forests, SVM, XGBoost) on drug response profiles of the CCLE database and fine-tuned the trained neural network models on 100 samples from the test dataset (GDSC database). We observed that, while fine-tuning can be beneficial for different models, it doesn't create an overall meaningful difference from the models that were not fine-tuned (Fig. 7A, see Supplementary Data 4 - Sheet 2 for paired bootstrap test statistics).

As the CCLE and GDSC databases have a relatively similar origin, resulting in good concordance with similar distributions, fine-tuning didn't yield a clear advantage. Therefore, we tested fine-tuning in a separate experiment where the source (training) dataset and target (test) datasets come from completely different sources. We built models to predict the cancer types of human tumor samples from three different TCGA cohorts (breast cancer, glioblastoma, and colorectal cancer) and used the trained model to predict the cancer types of cell lines derived from the corresponding three different cancer cell lines from the CCLE using gene expression and copy number variation data as input. We observed that all the models performed very poorly without fine-tuning, with an F1 score of ~0.16, with similar performances by Random Forest, XGBoost, and SVM models, too. However, fine-tuning the deep learning models using 50 samples led to a significant improvement in the prediction performance achieving F1 scores of up to 0.8 (Fig. 7B, see Supplementary Data 4 - Sheet 1 for paired bootstrap test statistics).

### Discovering biomarkers of drug response in cell lines
All model architectures implemented in Flexynesis, are equipped with a marker discovery module based on Integrated Gradients and

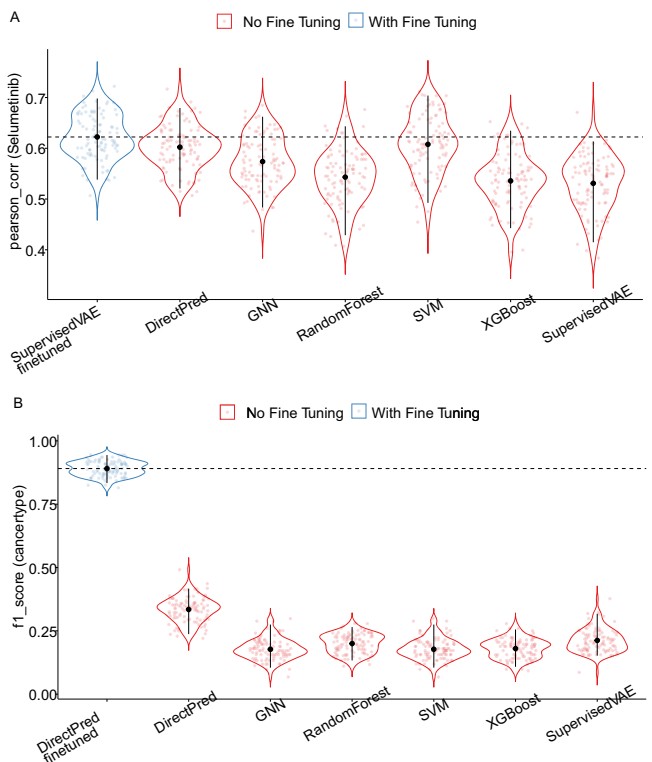

**Fig. 7 | Impact of model fine-tuning in prediction performance in datasets with similar data distributions (panel A: CCLE cell lines vs GDSC cell lines) and datasets with shifted data distributions (panel B: TCGA human tumor samples vs CCLE cell lines).** Violin plots show the distribution of performance values estimated from 100 bootstrap replicates. The solid black point indicates the mean, and the error bars represent the 95% confidence interval (CI) computed from the empirical bootstrap distribution. Individual bootstrap values are overlaid as jittered dots, color-coded by method or experimental condition. **A** Deep learning and classical machine learning methods were trained on the CCLE dataset for Selumetinib response levels (using gene expression and copy number variation data) and evaluated on the GDSC2 dataset. Deep learning models were further fine-tuned using 100 samples from GDSC2 and the models were evaluated on the remaining unseen test samples from the GDSC2 dataset. The best performing fine-tuned deep learning model is compared to other non-fine-tuned deep learning and baseline models. See also Supplementary Data 4 for paired two-sided bootstrap t-test statistics for additional drug response model performance comparisons with/without fine-tuning. **B** Models were trained on TCGA tumor samples from three different cancer types (breast cancer, glioblastoma, and colorectal cancer) and evaluated on CCLE cancer cells lines derived from the same three cancer types with and without fine-tuning the models using 50 samples during fine-tuning stage. The best performing fine-tuned deep learning model is compared to other non-fine-tuned deep learning and baseline models. See also Supplementary Data 4 for paired two-sided bootstrap t-test statistics. Source data are provided as a Source Data file.

GradientSHAP feature attribution methods[33–35]. In order to evaluate whether the trained Flexynesis models can capture known/expected markers, we constructed models predicting drug response, for eight drugs with known molecular targets. The models were trained on drug response data from CCLE and evaluated on the corresponding features from the GDSC dataset. We trained both a fully connected network (DirectPred), a supervised variational auto-encoder (supervised-vae), and graph-convolutional neural network (GNN-SAGE) using various data type combinations (mutations, mutations + RNA expression, and mutations + RNA expression + copy number variants). The top ten markers per drug were extracted from the best performing model among all the experiments (Fig. 8A) using both Integrated Gradients and GradientSHAP methods. The two methods yielded almost identical results (Supplementary Fig. 9), therefore we

report the feature attribution metrics only from the Integrated Gradients method. We labeled the top markers by data type and also by the presence of the marker in civicDB[36], a database of clinically actionable genetic biomarkers of drug response. For 6 out of 8 drugs, we could find at least one known marker, present in civicDB (Fig. 8B). In addition, we observe that the best performing models are never trained on "mutations-only". Top markers for each of the drugs are dominated by single nucleotide variants, however, we also observe that the best performing models (Fig. 8A) are the ones where the mutation data is complemented with at least the "RNA" layer, which is in line with previous findings, we and others have demonstrated before, that using the gene expression data on top of the mutation features significantly improves drug response prediction performance[37,38].

## The Flexynesis benchmarking pipeline

Previous benchmarking of different neural network architectures[13,14,39], showed that none of the methods outperform others in all tested scenarios. It is challenging to choose the best performing neural network architecture along with the type of multi-omic modalities best suited for a given task ahead of time. Additionally, it is possible that the accuracy of the classical machine learning methods, such as a random forest classifier, is sufficient for a given prediction task. Therefore, to attain the best performing model, we have to execute multiple experiments with different data type combinations, different fusion approaches, and different neural network architectures. Moreover, some tasks might benefit from building multi-task training, while others might perform better for the target variable of interest in a single-task setting.

To accommodate such combinatorial experimentations, we setup a benchmarking pipeline which can be configured to run different flavors of Flexynesis on different combinations of data modalities, different fusion options, fine-tuning options, along with a baseline performance evaluation using random forest, support vector machines, XGBoost and random survival forest methods. The pipeline then builds a dashboard with rankings of different experiments in terms of prediction performances for different tasks.

We ran the benchmarking pipeline on datasets with clinically relevant outcome variables and built a dashboard of rankings of the different experiments (See dashboard and Supplementary Data 6). We designed 14 different tasks across 5 different datasets in a total of 222 different experiments, where we tested different tools, tool flavors, data fusion and fine-tuning options. Immediate observation confirms previous findings that no single neural network model outperforms others in all tasks. Of the 14 tasks, the top ranking method was equally divided between deep learning models and classical machine learning models (SVM, Random Forest, XGBoost) (Fig. 9A, see Supplementary Data 6: Sheet 2 for paired bootstrap tests for the comparison of best performing deep learning and baseline models). Cross-experiment comparison of model performances suggest a slight edge for deep learning models (Fig. 9B) with small effect sizes, nonetheless. Furthermore, we compared the deep learning models in terms of omics data modality fusion options (see Methods: Data modality fusion options). Among the best performing models, we don't observe a significant difference between early or intermediate fusion settings (Fig. 9C). Similarly, fine tuned deep learning models don't show any insignificant improvement over the counterparts with no fine-tuning (Fig. 9D). Finally, among the GNN models, the choice of SAGE convolution method yields slightly better results in our experiments (Fig. 9E), but doesn't achieve statistical significance. These experiments suggest that the choice of deep learning versus baseline methods, or deep learning method settings such as model tuning, fusion options, or convolution methods probably depends on the specific task and it is not possible

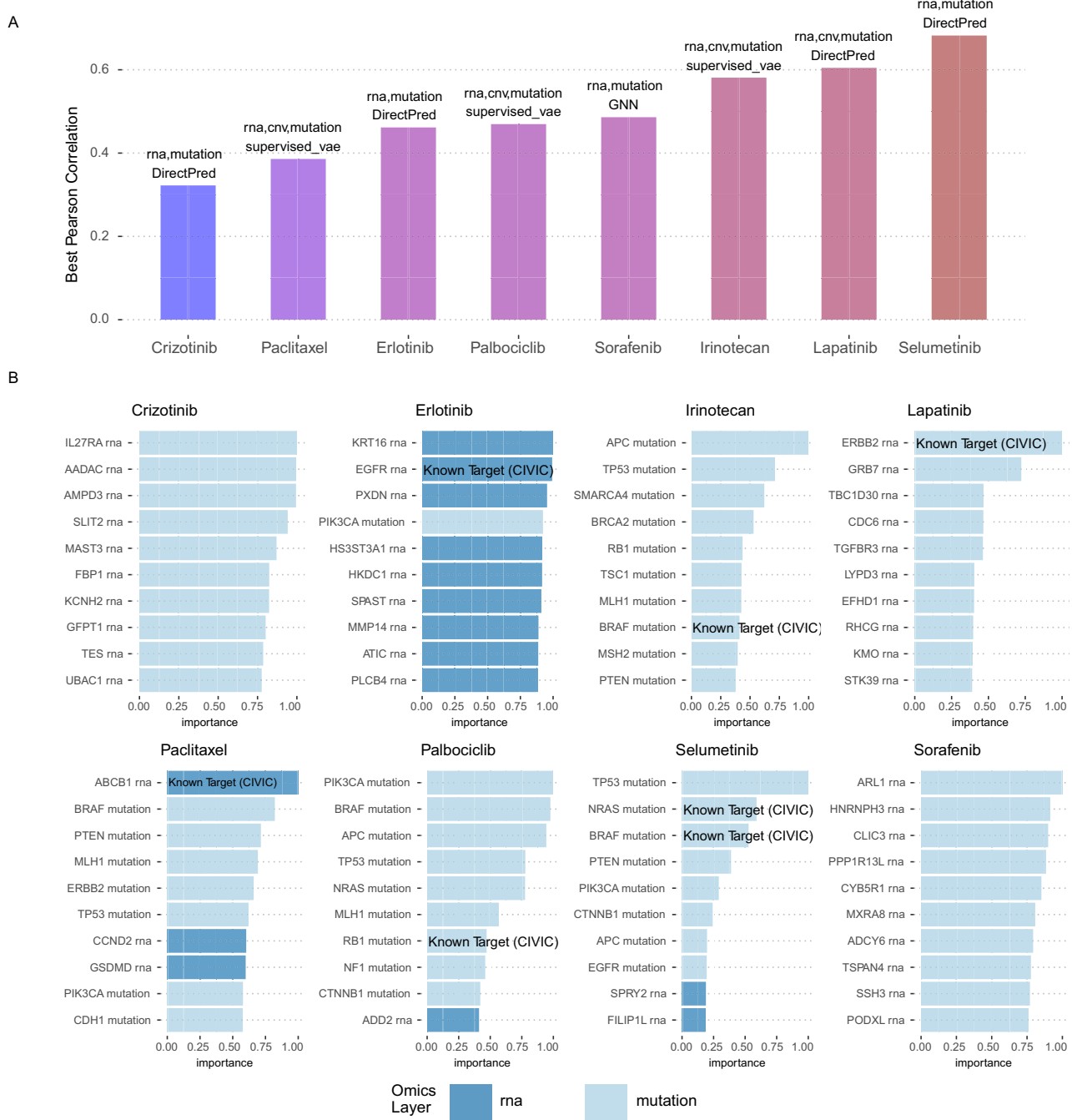

**Fig. 8 | Top markers discovered using the feature importance modules of Flexynesis in predicting the drug response values (trained on the CCLE dataset and evaluated on the GDSC2 dataset).** A fully connected network (DirectPred), a supervised variational autoencoder (supervised_vae), and a graph-convolutional network (GNN-SAGE) was trained on three combinations of data modalities: using only mutations; using mutations and RNA expression; using mutations, RNA expression, and copy number alterations. **A** Best performing model+data type combination for each drug is displayed. Color scale (shades of blue to red) reflects the pearson correlation score. **B** The top 10 markers (in the y-axis) discovered for each drug (based on the best performing model + data type combination depicted in panel (**A**). The markers are both labeled and colored by the corresponding data modality (dark blue: RNA expression, light blue: Mutation). The markers that are already known to be indicator markers for the corresponding drug according to the CIViC (Clinical Interpretation of Variants in Cancer) database are labeled as "Known Target (CIVIC)". The x-axis displays the relative importance of the top markers, where the best marker has a value of 1. While most drugs have dominantly mutation markers in the top 10, the best performing models always have RNA expression as an additional data modality. Source data are provided as a Source Data file.

to generalize to all possible situations. The value of different approaches is task specific, therefore we advise running multiple experiments to obtain the best model for the dataset at hand. This accessory pipeline ameliorates the execution of such experiments. The pipeline is available at https://github.com/BIMSBbioinfo/flexynesis-benchmarks.

## Discussion
In this paper, we presented Flexynesis, a deep learning based bulk multi-omics integration suite with a focus on (pre-)clinical variable prediction. Despite the availability of many published deep learning-based methods, the main reason for developing this package was to provide an improved user experience when adapting deep learning

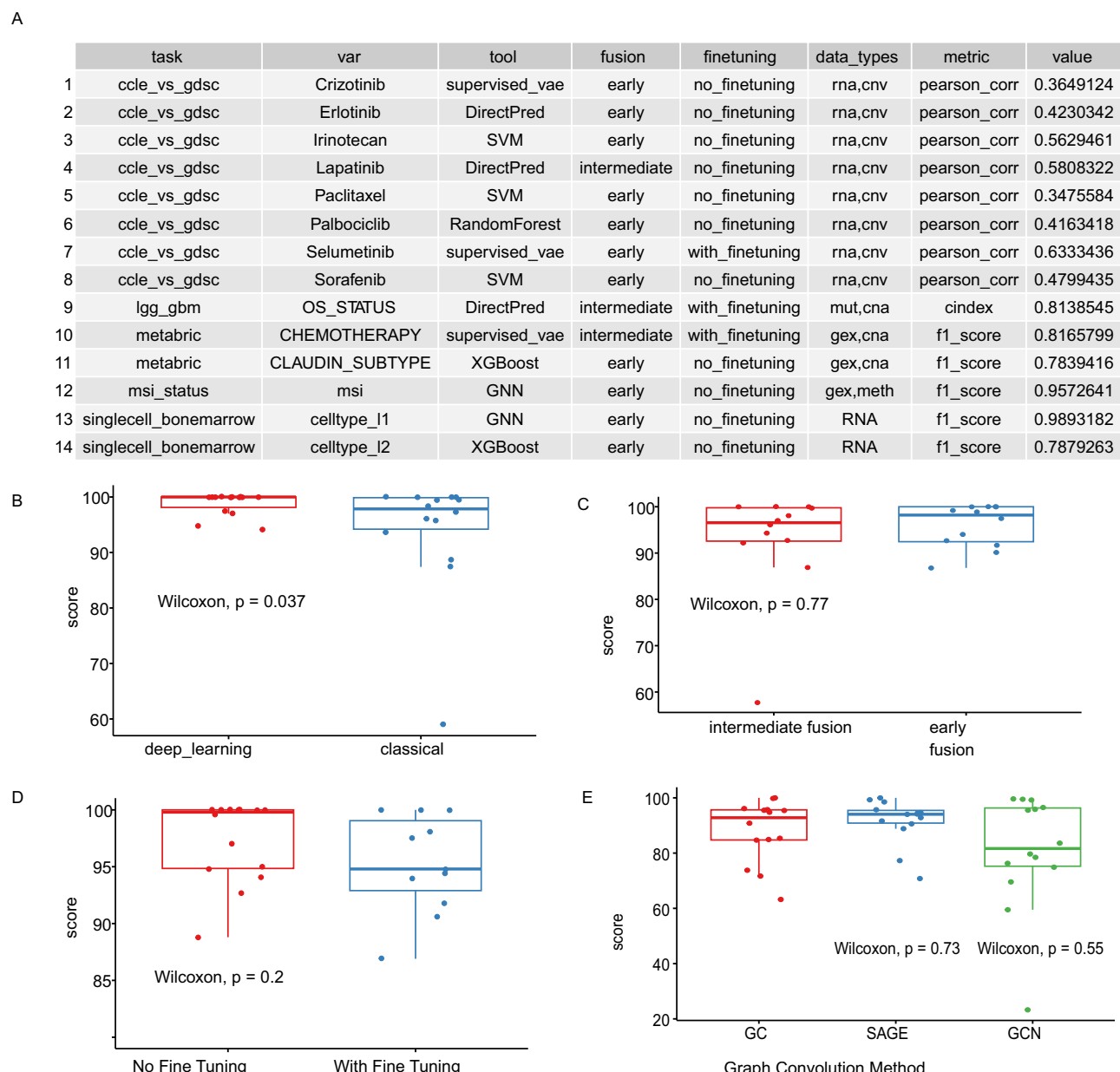

**Fig. 9 | Summary of benchmarking results using different tools and data integration options with the Flexynesis benchmarking pipeline.** The "score" in the y-axis for panels B-E are scaled to 100 (where the best model gets a score of 100) in order to enable comparison of different scoring metrics. **Description of box-plots:** the center line indicates the median, the box bounds represent the 25th and 75th percentiles, and the whiskers extend to the most extreme values within 1.5 × the interquartile range (IQR). Individual data points are overlaid using jitter to visualize sample density and spread. Statistical comparisons between groups were performed using a two-sided Wilcoxon test. **A** Best performing model setup for each of 14 different tasks. **B** Comparison of the best result of among any deep learning method ($N = 14$) with the best result of the classical machine learning approaches ($N = 14$) (two-sided Wilcoxon test, $p = 0.037$) **C** Comparison of the best performing data fusion strategies: intermediate fusion ($N = 12$) and early fusion ($N = 12$) for the deep learning models in experiments where there were at least 2 data modalities (two-sided Wilcoxon test, $p = 0.77$). **D** Comparison of the best performing model with ($N = 11$) and without ($N = 14$) the fine-tuning option (two-sided Wilcoxon test, $p = 0.2$). **E** Comparison of the best performing GNN models in terms of different convolution options: SAGE ($N = 14$) vs GC ($N = 14$) (two-sided Wilcoxon test, $p = 0.73$) and GCN ($N = 14$) vs GC ($N = 14$) (two-sided Wilcoxon test, $p = 0.55$). Source data are provided as a Source Data file. See Supplementary Data 6 for full table of results and bootstrap testing statistics for comparisons of best performing deep learning and baseline models.

for multi-omic data analysis. Existing methods lack one or more of the important components, where the absence of any of these components creates significant overhead for the users when adapting deep learning applications in their experiments. We provide a package that is easily installable, supported with good documentation, real-life benchmarking datasets with example applications, and automates data cleanup and harmonization, feature selection, hyperparameter optimisation, model evaluation, and feature importance ranking. The package is designed in a way that the user can easily switch different kinds of model architectures, can easily decide which data types to use in modeling by simply providing a list of files, experiment with modeling different kinds of clinical variables in single-task or multi-task settings and build models for supervised (regression, classification, survival), unsupervised, or cross-modality tasks without having any in-depth experience in building deep learning architectures. Thus, the user can focus on the biological

context of the study and come up with interesting questions to solve with this diverse toolkit.

It has been previously shown that deep learning may struggle to outperform classical machine learning methods[15–17], which we have also observed in a subset of our benchmarking experiments. Even though classical machine learning methods might perform as good as a neural network in certain situations, the decision to use deep learning is not guided solely by the prediction performance for a given task. Deep learning offers a broader level of flexibility such as tolerance for missing labels, support for multi-task modeling, enables both supervised/unsupervised/and matrix-to-matrix predictions while simultaneously allowing dimension reduction. Furthermore, pre-trained deep learning models can be fine-tuned on a separate dataset enabling transfer learning. Finally, deep learning gains a competitive advantage with increasing amounts of data[39,40], which should be more commonplace in clinical research as multi-omic profiling becomes easier and cheaper over time.

Although we use Flexynesis mainly for multi-omics data integration, current implementation is built in a data-agnostic manner, where the only assumption is that the input data matrices are in a tabular format, in other words, it is a multi-modal data integration tool suite. Along this line of thought, we have utilized not just bulk omics data, but other kinds of tabular data such as protein-language model embeddings. With the same motivation, we also wanted to observe if it would work on supervised tasks on single-cell multi-omics, and it turned out to be useful for the cell type classification task using CITE-Seq data (Fig. 9A, Supplementary Data 6). However, we don't see Flexynesis in its current form as an alternative to single-cell-oriented tools. A major difference in the kinds of applications where we utilize Flexynesis in contrast to the typical single-cell pipelines, is that there is a big emphasis on supervised tasks in (pre)-clinical cohort studies as there are always some clinical sample labels available that can guide the analysis, such as survival outcomes, disease subtypes, histology, patient characteristics such as age, gender, so on and so forth. On the other hand, the single-cell omics applications are usually driven by unsupervised applications, where individual cell identities are not always apparent, therefore unsupervised approaches for data integration and clustering is typically carried out, which is followed by differential marker analysis to ascribe identities to such clusters. Although we have also provided such functionalities for Flexynesis (unsupervised clustering using variational autoencoders, clustering, and associated utility functions for visualisation), purely unsupervised approaches are not our main goal, rather a side-product of what we would like to achieve with Flexynesis. Other single-cell oriented tools are better equipped for investigation of such unsupervised clusters with methods for unsupervised integration, marker analysis, and a variety of accessory utilities one needs to inspect such clusters of single-cells. As per the usual unsupervised clustering that are usually carried out for the (pre-)clinical bulk sequencing cohorts such as disease subtyping, we expect Flexynesis to be used in a supervised manner. For instance, to investigate prognostic disease subtypes, a model can be trained with supervisor MLPs that predict survival outcomes or MLPs that predict treatment outcomes. Thus the sample embeddings would reflect clusters that are guided by prognostic labels. Similar ideas can be applied for any subtyping scheme, whether diagnostic or prognostic. Thanks to the multitasking support, one can do clustering of samples across multiple patient covariates, thus delineating intersectional diagnostic/prognostic subtypes. So, even for the typically unsupervised tasks such as disease subtyping, we expect Flexynesis to be used in conjunction with some supervision associated with clinical variables.

By easily adapting Flexynesis into a bioinformatics pipeline, we have assessed both the relative performance of different flavors of deep learning architectures, along with other parametric choices one can make in multi-omics integration such as the combination of data modalities, different fusion options, and fine-tuning options. The benchmarking pipeline we built with various real-life datasets should allow both the developers in assessing the strengths and weaknesses of the novel features contributed to the package, but also guide the users to make choices based on the nature of the modeling task.

For future development of the toolkit, each of the multiple components will be easily expanded by implementing alternative methods. Currently we offer multiple alternative models for training and methods for marker discovery, but not for feature selection or hyperparameter optimisation. We plan to implement alternative hyperparameter optimization algorithms provided by libraries such as Ray Tune[41] or Optuna[42], and expand the marker discovery, using various ranking algorithms available within the Captum library. Feature selection can be extended using unsupervised feature selection methods such as Fractal Autoencoders[43].

As a final remark, it is important to note that what we developed here is not a set of novel deep learning algorithms. None of the components we built are novel, however the innovation comes from how these components are brought together into a usable package. Flexynesis improves user experience and makes multi-omic deep learning accessible to a broader audience.

## Methods

Flexynesis is a pytorch-lightning based deep learning framework designed for bulk multi-omics data integration with a focus on precision oncology applications, however it is possible to use it for any tabular multi-modal datasets. Flexynesis workflow consists of the following main steps: importing the multi-omics data and metadata for training and testing samples, running a bayesian sequential hyperparameter optimisation routine using scikit-optimize package[44] on the training dataset and choosing the best model parameters in terms of validation metrics, evaluating the best performing model on the test (holdout) dataset, and computing the ranking of the input features in terms of importance using the Captum package[34]. If the user opts for a fine-tuning procedure, the trained model is fine-tuned on a subsample of the testing set and the fine-tuned model is evaluated on the remaining test samples.

### Importing the training and test datasets

Flexynesis expects a path to a data folder which contains training and testing data. Both training and testing data folders should contain at least one matching data modality (e.g. omics1.csv, omics2.csv …) as a data matrix and a meta-data file that contains sample labels for each sample (clin.csv). The omics data files contain omic profiles of samples where the column names represent unique sample/patient ids and row names represent the profiled omic features. The sample metadata file (clin.csv) contains the unique sample names in rows and clinical features (outcome variables) as column names.

During the data import (Supplementary Fig. 1), Flexynesis checks for common file format errors and cross-checks information available in omics data files and metadata files to make sure that both training and testing datasets are ready for downstream analysis. After the sanity checks, the training data is further processed. Common issues with tabular data such as missing values are imputed, features with low variance are removed, samples with no available features are dropped. After the data cleanup, depending on the user's requirements, a feature selection is implemented to keep the top most informative features based on the Laplacian Scoring method[45]. Among the top most informative features, highly redundant features are also dropped to keep unique and informative features. The feature selection is done for each data modality separately. The user can choose to keep a minimum number of features per data modality. In case the user opted to use a graph convolutional network, the genetic interaction networks are downloaded from the STRING database (according to the

**Table 1 | Hyperparameter Optimization Search Spaces**

| Model | Hyperparameter | Type | Range / Values | Notes |
|---|---|---|---|---|
| DirectPred / supervised_vae / CrossModalPred / MultiTripletNetwork | latent_dim | Integer | 16–128 | – |
| | hidden_dim_factor | Real | 0.2–0.5 | Relative to input dimension |
| | lr | Real (log) | 1e-4–1e-2 | Learning rate |
| | supervisor_hidden_dim | Integer | 8–32 | – |
| | max epochs | Categorical | [500] | Fixed |
| | batch_size | Categorical | 32–128 (powers of 2) | Dataset-dependent |
| GNN | latent_dim | Integer | 16–128 | – |
| | node_embedding_dim | Integer | 4–32 | Node embedding size |
| | num_convs | Integer | 1–4 | Number of convolutional layers |
| | lr | Real (log) | 1e-4–1e-2 | – |
| | supervisor_hidden_dim | Integer | 8–32 | – |
| | max epochs | Categorical | [500] | Fixed |
| | activation | Categorical | ['relu'] | Fixed |
| | batch_size | Categorical | 32–128 (powers of 2) | Dataset-dependent |

requested organism id) and the training data modalities are filtered to keep only the features that are found in the interaction networks.

After feature selection, the training data is scaled and centered and optionally log-transformed. Once the modifications to the training data are finished, testing data is harmonized with the training data to make it compatible with the final model. To avoid data leakage, testing data is only scaled/centered using the scaling factors learned from the training data and the features selected for training data are kept in the testing data. Thus the testing data does not influence feature selection or data normalization. All omic data and sample labels are finally converted into pytorch tensors.

If the user decides to use a subset of the clinical variables as covariates in the model, the variables are processed to convert categorical variables into numerical variables by one-hot-encoding. The numerical variables are kept as they are. Missing features are imputed to the median values. Thus the list of covariates are converted into a numerical matrix which is used as an additional input data modality for model training.

## Hyperparameter optimisation

In the current implementation of the Flexynesis package, a Bayesian sequential hyperparameter optimization procedure is followed (Supplementary Fig. 2). Initially a random set of model specific hyperparameters are assigned and scikit-optimize package[44] is used to suggest different parameters after each hyperparameter optimization iteration. The user decides on how many iterations to carry out. The commonly optimized hyperparameters are "latent_dim": the number of units to use for the encoding (the number of dimensions to aim for the sample embeddings) per data modality, "hidden_dim_factor": the size of the hidden layer units in relation to the size of the previous network layer. Instead of setting this to absolute value terms, we decided to make it into relative values so that the parameter search behaves similarly depending on different input sizes, as different data modalities may have different number of features, thus having different input layer sizes in the network. "Supervisor_hidden_dim": represents the number of units to use in the hidden layer of the MLP heads. The input layer size of this MLP is the total size of the latent factors (number of modalities x latent dim parameter). "lr": the learning rate for the ADAM optimiser[46], "epochs": the max number of epochs to continue the training. We use the 'early_stop_patience' callback so that the training is stopped if the validation loss values aren't improving after a set number of epochs. Using the early stop patience significantly improves training speed and also avoids overfitting on the training data, thus improving model generalization on test data. See

Table 1 for the default search space configurations used by different model architectures.

## Model/network/encoding options

Flexynesis currently contains a selection of architectures which can be used to train the models.

- **DirectPred**: A multi-task fully connected neural network for direct prediction of one or more target variables (Supplementary Fig. 4).
- **supervised_vae**: A variational autoencoder model architecture with MMD loss (Supplementary Fig. 5).
- **CrossModalPred**: A cross-modality encoder/predictor, which is a special implementation of the variational auto-encoders, in which the input data modalities and output data modalities can be set to different subsets of the available data modalities (Supplementary Fig. 6).
- **MultiTripletNetwork**: A fully connected neural network implemented with a triplet loss-based contrastive learning (Supplementary Fig. 7).
- **GNN**: A graph neural network that by default uses the STRING database as interaction networks. Different graph convolution options are available: GraphConv[47], GCNConv[48], and SAGEConv[49]. Currently supports only early-fusion of data modalities (Supplementary Fig. 8).

All of these networks can be augmented with one or more Multi-Layered-Perceptrons (MLPs) depending on the number of target variables the user wants to build a prediction model for. The user can select one or more target variables for regression/classification tasks. On top of these regression/classification heads, a survival MLP can be added for which the user needs to provide two variables, where time represents the time since last followup, and event is a binary value (0 or 1) which represents whether an event has occurred since the last follow up. An event can be any clinically relevant event such as disease progression or a death event.

## Runtimes and resources

In order to highlight resources consumed by a typical Flexynesis run, we ran a resource profiling experiment, in which we used 500 breast cancer samples with gene expression and copy number alteration data modalities consisting of 2000 features each. We profiled the time (wall clock time) it takes to import data and run a single hyperparameter optimization step. We also profiled the CPU (Intel(R) Xeon(R) Platinum 8168 CPU @ 2.70 GHz) RAM usage and GPU (Nvidia Tesla P40) RAM usage statistics for 5 different neural network architectures with two

different fusion approaches (early/intermediate). We observe that DirectPred (classical feed forward network) with intermediate fusion utilized with GPU is the fastest option. Models with intermediate fusion have smaller number of parameters, therefore faster to train and consume less memory (Supplementary Fig. 10, Supplementary Data 7).

## Data modality fusion options

Flexynesis supports two kinds of data modality fusion options for fusing the omics layers. With "early fusion", all input omic matrices are concatenated prior to training. With intermediate fusion, the input omic matrices are individually propagated through dedicated encoding networks. The output layer of the encoding networks (or the latent layer in the auto-encoder architectures) are concatenated and used as input to the MLP heads for each target/survival variable.

## Model training and loss functions

During model training, the training data is split by default into 80/20 portions for training and validation. The user can also select to do a k-fold cross-validation, in which the training data will be split into k-folds. For each MLP head dedicated to the corresponding outcome variable, a loss function is computed according to the variable types. If the variable is a continuous/numeric variable, a mean-squared-error loss is computed. If the variable is a categorical variable, a cross-entropy loss value is computed. If the variable is a survival variable, the cox-proportional hazards loss function is computed. The VAE models have an additional loss value: Maximum Mean Discrepancy (MMD) Loss[25]. The MultiTripletNetwork models use a triplet loss for contrastive learning, where the similarity between the anchor sample and positive examples are maximized, while the similarity between the anchor sample and the negative examples are minimized[50].

Depending on the model architecture and the number of MLP heads, there may be multiple loss values computed for a training task. The total loss is computed by summing up the individual loss values. However, as different loss functions can have different scales, it may be beneficial or even necessary to have a weighting schema to avoid one of the loss values to dominate the training. For this, we implemented the uncertainty weighting method[51], which can be disabled.

The total validation loss guides the training process. The final validation loss obtained from the training run is used to inform the hyperparameter optimiser to set the next set of hyperparameters for the next run.

## Model fine-tuning

When the user opts for a fine-tuning procedure, a portion of the test samples (user defined) are used to fine-tune the trained model parameters (Supplementary Fig. 3). The fine-tuning can be beneficial in cases of dramatic shifts in dataset distributions between training and test datasets. The fine-tuning procedure consists of a five-fold cross-validation scheme on a grid searching a combination of different learning rates and different model parameter freezing strategies (freeze the encoders, freeze the MLP heads or freeze none). Again an early stop callback is used with a low patience (3 epochs) to avoid overfitting to the testing dataset. The best model from this cross-validation scheme is chosen as the fine tuned model to be evaluated on the remaining test samples.

## Model performance evaluation metrics

Once a model is optimized on the training/validation sets, the model is evaluated on the testing dataset. For regression tasks, we compute the mean squared error, R-squared, and the Pearson correlation coefficients to evaluate the performance of a model. For classification tasks, we compute the balanced accuracy, F1 score (weighted average), AUROC (weighted average), AUPR (weighted average), and kappa statistic. For survival tasks, we compute the Harrel's C-index as the model evaluation metric.

## Feature importance calculation for marker detection

After model training, most important features for each target variable and for each factor within the target variable are calculated using two alternative feature attribution methods: the Integrated Gradients method[33] and GradientSHAP method.

## Assessment of baseline performance

Flexynesis also contains functions to evaluate the prediction performance of classical machine learning algorithms on the same task. For regression and classification tasks, random forests[52], support vector machines[53], and XGBoost[54]; for survival tasks, random survival forests[55] are utilized. These models are trained using a 5-fold cross-validation scheme where hyperparameter optimization is carried out on the training data and the best performing model is evaluated on the test dataset with the same metrics as we use for the neural network models. Scikit-learn library was extensively utilized for these methods and computing the evaluation metrics[56].

## Paired bootstrap evaluation

We used paired bootstrapping to compare model performance across multiple metrics. For each task, 100 bootstrap samples were generated by resampling test instances with replacement. Model predictions were evaluated using task-specific metrics (e.g., F1 score, Pearson correlation, concordance index), applied to each bootstrap sample. Confidence intervals (95%) were computed using the percentile method. For pairwise model comparison, we conducted a paired $t$-test over bootstrap scores.

## Network analysis

Human genetic interaction networks were downloaded from the STRING database[31] and network centrality measure (hubness score) was calculated using the igraph R package[57].

## Batch-Alignment of Embeddings Post-training

In the cases when different datasets were combined to build models, sample embeddings originating from different datasets can be aligned in the embedding space after the model training phase. Two different approaches are currently implemented:

1.  Batch alignment using reciprocal PCA with mutual nearest neighbors: We utilize Reciprocal PCA (rPCA) to align sample embeddings derived from two different datasets by applying Principal Component Analysis (PCA) independently to each batch and projecting the data into the other's principal component space. Anchor samples are identified using Mutual Nearest Neighbors (MNN) where the anchors represent samples that are closest across batches in both transformed spaces. Using the anchors a reciprocal PCA space is learned and other non-anchor samples are transformed in this space[58].
2.  Batch alignment using optimal transport: we use Python POT library[59] to compute optimal transport plans between two batches of sample embeddings, where we map the samples from one batch to another in a way that preserves their relative Euclidean distances, harmonizing batch-specific variations.

## Datasets

**CCLE.** Multi-omic and drug response data for the cell lines from the CCLE[10] was downloaded from https://zenodo.org/records/3905462 and processed using the PharmacoGx R package[60].

**GDSC2.** Multi-omic and drug response data for the cell lines from the GDSC was downloaded from https://zenodo.org/record/3905481 and processed using the PharmacoGx R package[60].

**Datasets from Cbioportal.** The merged cohorts for Lower Grade Glioma (LGG) and Glioblastoma MultiForme (GBM) dataset[21] were

downloaded from: https://www.cbioportal.org/study/summary?id=lgggbm_tcga_pub.

The TCGA pan-cancer atlas datasets used in the fine-tuning use-case (Fig. 6B), including multi-omics data for colorectal cancer[61], breast invasive carcinoma[62], and glioblastoma multiforme[21] was downloaded from:

– https://www.cbioportal.org/study/summary?id=coadread_tcga_pan_can_atlas_2018
– https://www.cbioportal.org/study/summary?id=brca_tcga_pan_can_atlas_2018
– https://www.cbioportal.org/study/summary?id=gbm_tcga_pan_can_atlas_2018

**METABRIC.** Multi-omic data for the metastatic breast cancer cohort from the METABRIC study[23] was downloaded from Cbioportal:

https://www.cbioportal.org/study/summary?id=brca_metabric

**Single-cell CITE-Seq of bone marrow.** Single-cell CITE-Seq dataset[63] was downloaded and processed using Seurat (v5.1.0)[64]. 5000 cells were randomly sampled for training and 5000 cells were sampled for testing.

**DepMap.** The omics data, CRISPR screens and PRISM drug screening data for cell lines from the DepMap project[26] was downloaded from the DepMap Portal (https://DepMap.org/portal).

**TCGA data.** The TCGA datasets were downloaded using the TCGA-Biolinks package[65].

**Prot-trans sequence embeddings.** Protein sequence embeddings for each gene was obtained using the prot_t5_xl_uniref50 transformer model (available at https://huggingface.co/Rostlab/prot_t5_xl_uniref50)[29]. The canonical protein sequences of the human proteome were downloaded from the UniProt database[66]. For each protein sequence, the transformer model outputs a numeric matrix of the dimensions 1024 x N where N is the number of amino-acids in the protein sequence. For each protein sequence, the row-wise averages were calculated to obtain protein-level 1024 dimensional vector embeddings.

**describePROT.** Structural/functional features of human protein sequences were downloaded from the describePROT database[30]:

http://biomine.cs.vcu.edu/servers/DESCRIBEPROT/download_database_value/9606_value.csv.

### Data visualization
Data visualization methods implemented in the Flexynesis package uses Matplotlib[67] and Seaborn[68] and lifelines[69] python libraries.

We used icons from www.flaticon.com in the graphical abstract of this manuscript.

### Clustering
Flexynesis is equipped with utility functions to cluster a given matrix and choose optimal clusters. Currently two clustering methods are supported: Louvain clustering from the community package[70] and k-means algorithm from the scikit-learn package[56], where the clustering can be done for different values of k and the optimal clustering result can be selected by Silhouette score rankings.

### Integration with the Galaxy Server
Flexynesis was integrated into the Galaxy Server deployed into the European Galaxy platform (usegalaxy.eu) to improve accessibility and usability[71]. To guarantee reproducibility and ease of installation in the Galaxy environment, a Bioconda package was developed[72], and a Bio-Container was created[73]. Thanks to this integration, users without access to large computing resources may run Flexynesis without requiring command-line expertise using an intuitive online graphical interface. Furthermore, Flexynesis can be integrated within established Galaxy workflows, such as RNA-seq and ATAC-seq, to enhance reproducibility, and ease of use, and enable broader adoption among the scientific community. In-depth testing and documentation were conducted to provide the optimal performance and user experience inside the Galaxy environment.

### Statistics and reproducibility
In this manuscript, the primary source of data was publicly available multi-omic datasets of patient cohorts and cancer cell lines. No statistical method was used to predetermine sample size. No data were excluded from the analyses. The experiments were not randomized. The Investigators were not blinded to allocation during experiments and outcome assessment.

### Reporting summary
Further information on research design is available in the Nature Portfolio Reporting Summary linked to this article.

## Data availability
All the datasets used in this study are previously published datasets (see Methods for how they were further processed for training). None of these datasets are under restricted accesss: - Multi-omic and drug response data for the cell lines from the CCLE[10] can be downloaded from https://zenodo.org/records/3905462. - Multi-omic and drug response data for the cell lines from the GDSC can be downloaded from https://zenodo.org/record/3905481. - The merged cohorts for Lower Grade Glioma (LGG) and Glioblastoma MultiForme (GBM) dataset[21] are available at: https://www.cbioportal.org/study/summary?id=lgggbm_tcga_pub. - Multi-omics data for colorectal cancer (TCGA) are available at Cbioportal: https://www.cbioportal.org/study/summary?id=coadread_tcga_pan_can_atlas_2018. - Multi-omics data for breast invasive carcinoma (TCGA) are available at Cbioportal: https://www.cbioportal.org/study/summary?id=brca_tcga_pan_can_atlas_2018. - Multi-omics data for glioblastoma multiforme[21] are available at Cbioportal: https://www.cbioportal.org/study/summary?id=gbm_tcga_pan_can_atlas_2018. - Multi-omic data for the metastatic breast cancer cohort from the METABRIC study[23] are available at Cbioportal: https://www.cbioportal.org/study/summary?id=brca_metabric - Single-cell CITE-Seq dataset[63] is available via the Seurat (v5.1.0)[64] package. - The omics data, CRISPR screens and PRISM drug screening data for cell lines from the DepMap project[26] is available at the DepMap Portal (https://DepMap.org/portal). - The TCGA datasets are available via the TCGABiolinks package[65]. - Structural/functional features of human protein sequences are available at the describePROT database[30]: http://biomine.cs.vcu.edu/servers/DESCRIBEPROT/download_database_value/9606_value.csv. All the above-mentioned datasets were further processed to enable training Flexynesis models. The datasets prepared for training Flexynesis models and the outputs of the model training required to reproduce the figures and tables in this manuscript are available in the Zenodo database under DOI accession 10.5281/zenodo.16442997 (https://zenodo.org/records/16442998)[74]. The source data for the figures are provided with this paper. Source data are provided with this paper.

## Code availability
The source code repositories developed for this study at the time of this manuscript submission are deposited at the Zenodo database: - The raw/processed dataset descriptions along with code to reproduce manuscript figures can be found at https://github.com/BIMSBbioinfo/flexynesis_manuscript. (v.1.0.3 is available at https://zenodo.org/records/16444303)[75]. The repo is accessible with an MIT licence. - The core Flexynesis software package is available at: https://github.

com/BIMSBbioinfo/flexynesis (v1.0.0 available at https://zenodo.org/records/16444460)[76]. The repo is accessible with a Modified MIT License for Academic and Non-Commercial Use. - The accessory benchmarking pipeline utilizing Flexynesis is available at: https://github.com/BIMSBbioinfo/flexynesis-benchmarks (v1.0.0 is available at https://zenodo.org/records/16443659)[77]. The repo is accessible with a Modified MIT License for Academic and Non-Commercial Use.

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

## Acknowledgements

We are thankful to Aaron Kollotzek, Alexander Blume, Artur Manukyan, Fabian Janosch Krueger, Jacqueline Jansen, and Jonas Freimuth for joining the user experience hackathon and testing the software. We also thank Dan Munteanu and Martin Siegert for their help with the compute servers. A.A. received additional support from the TEP-CC consortium (Bruno and Helene Jöster Foundation) and the DFG CRCs 1588. The work by A.N.N and B.G was was funded by the German Federal Ministry of Research, Technology, and Space BMFTR grant 031 A538A de.NBI-RBC and the Ministry of Science, Research and the Arts Baden-Württemberg (MWK) within the framework of LIBIS/de.NBI Freiburg. The results published here are in whole or part based upon data generated by the TCGA Research Network: https://www.cancer.gov/tcga. The results published here are in whole or part based upon data generated by the Therapeutically Applicable Research to Generate Effective Treatments (https://www.cancer.gov/ccg/research/genome-sequencing/target) initiative, phs000218. The data used for this analysis are available at the Genomic Data Commons (https://portal.gdc.cancer.gov).

## Author contributions

B.U. co-conceived the project, implemented the Flexynesis software and the benchmarking pipeline, designed the experiments, carried out data analyses, wrote the manuscript, and supervised the software contributors. T.S. contributed to the GNN model architecture development. A.S. helped with adding different graph convolution options. R.W. helped package the software for pypi and Guix. M.M.S. designed the dashboard of the benchmarking pipeline results. A.N.N built the Bioconda package and integration tool for the Galaxy server with supervision from B.G. V.F. contributed to the conception of the project, helped with interpretation of results, and edited the manuscript. A.A. co-conceived the project, helped with data interpretation and designing experiments, edited the manuscript and provided funding and resources.

## Funding

## Competing interests

Bora Uyar, Vedran Franke and Altuna Akalin are part of a related patent application (Method for analysis of omics data, US Patent App. 18/274,271, 2024). The status of the patent application is currently pending. The patent application is about unsupervised integration of multi-omic datasets using neural networks (primarily auto-encoders). The current manuscript expands on the ideas listed in the patent by inclusion of a diversity of neural network architectures with a focus on supervised learning strategies, rather than only unsupervised learning. The remaining authors declare no competing interests.
