## [Peer Review File · Nature Communications]

Flexynesis: A deep learning toolkit for bulk multi-omics data integration for precision oncology and beyond

Corresponding Author: Dr Altuna Akalin

Version 0:

Reviewer comments:

Reviewer #1

(Remarks to the Author)

In this study, Bora et al. introduced Flexynesis, a comprehensive solution designed to enhance the utility and applicability of deep learning in multi-omics data analysis. Flexynesis streamlines data processing, enforces structured data splitting, and ensures rigorous model evaluation. This toolset could make deep-learning based bulk multi-omics data integration in the context of clinical/pre-clinical data analysis and marker discovery more accessible to a wider audience with or without experience in deep-learning development. The authors demonstrated the versatility of Flexynesis through various use cases, including drug response prediction, cancer subtype modeling, survival analysis, and biomarker discovery. Indeed, methods for bulk multi-omics data integration are important, and this toolset could make deep learning-based multi-omics data analysis more accessible to wider users. So, this study is timely and very useful for the community. However, there are several technical limitations. Some issues are itemized below.

Major points:

1. The article's title mentions bulk multi-omics data integration, but subsequent case studies utilize single-cell data. Can the current pipeline be extended to single-cell multi-omics data integration? What are the limitations of the current pipeline in integrating single-cell multi-omics data?
2. Compared to single-task modeling, does multi-task modeling, while capable of predicting multiple outcome variables simultaneously, result in decreased performance for one specific outcome variable?
3. Could you create a simple diagram of the omics data preprocessing steps to facilitate reader understanding?
4. How can the impact of batch effects be assessed and mitigated when integrating data from different batches or sources?
5. It will be helpful to provide an approximate description of the hardware used and the runtime.
6. Given the excellent performance of XGBoost on tabular data, it is recommended to include it as a baseline.
7. In the "Marker discovery" section, the title does not reflect the evaluation of feature importance. It is recommended to clearly specify the model's functions and applications in the title. Moreover, feature importance evaluation is often unstable; could additional methods besides IG, such as SHAP, be included to assess whether the results are consistent with IG?
8. The test cases do not include GNN models; however, GNN methods have shown certain advantages in previous studies. It is recommended to incorporate GNN models in some case studies.
9. In "Cross-modality learning" section, "In addition, we attached a supervisor MLP to guide the network to predict the hubness-score of each gene in the genetic interaction networks obtained from the STRING database assuming that the centrality of a gene in biological interaction networks could be a contributing factor in its essentiality for cell survival." An ablation experiment is needed to demonstrate that the inclusion of the supervisor MLP indeed aids in reconstructing the cancer cell line dependency scores.

10. "During model training, the training data is split by default into 80/20 portions for training and validation. The user can also select to do a k-fold cross-validation, in which the training data will be split into k-folds."

Could an independent test set be split for evaluating model performance, such as an 80/10/10 split? and then use k-fold cross-validation for hyperparameter optimization, use independent test set for evaluation.

Minor points:

1. For some of the more complex model setups, such as cross-modality learning, could you provide a simple model architecture diagram to facilitate reader comprehension?

2. Why were AUC and AUPR not used in the evaluations of classification tasks?

3. In "Improving model performance via model fine-tuning" section, what is the specific architecture of the supervised-VAE model?

"We observed that, while fine-tuning does not guarantee an improvement in model performance it can provide a boost in accuracy (Figure 6A)." This sentence is confusing.

4. In figure 3A, using different background colors for high-risk and low-risk groups would be more effective for presentation.

5. In figure 4C, what does the colors in the bar represent?

6. There is significant room for improvement in the visualization of the figures.

(Remarks on code availability)

Reviewer #3

(Remarks to the Author)

The manuscript presents Flexynesis, a deep learning pipeline aimed at integrating bulk multi-omics data, enhancing usability and adaptability for various precision oncology applications. The authors emphasize the limitations of existing methods, such as lack of transparency, modularity, and narrow applicability, which Flexynesis seeks to address. They demonstrate the framework's capabilities across various modeling tasks including regression, classification, survival analysis, and biomarker discovery, along with a benchmarking pipeline for model performance evaluation. Overall, the novelty of the underlying algorithms is very limited, with an emphasis on existing tool integration and optimization rather than development of new methods. It lacks sufficient explanations for method selection and more.

1. The Introduction section needs to be revised to make it sharp and concise. The logic and core message are not clear.

2. While the manuscript outlines the importance of multi-omics data integration in precision oncology, it could strengthen its case by providing more detailed examples of how Flexynesis has improved real-world clinical outcomes or research findings. What specific applications have demonstrated significant advancements due to this tool?

3. The true innovation of this work is unclear. No new methods, scores, or standards is proposed and show superior performance than existing ones.

4. The authors state that Flexynesis supports various architectures, yet there is a lack of comparative analysis on why certain architectures were chosen over others for specific tasks. Also, is Flexynesis flexible to include any new models developed in the future? How?

5. It is not clear the output of Flexynesis. One of the gaps Flexynesis wants to fill is the limited tasks. However, Flexynesis also only focuses on five tasks of regression, classification, survival, clustering, and cross-modality prediction. If the author considers Flexynesis as an all-in-one or one-stop tool for bulk multi-omics, then, more analyses should be included, especially those omics-specific tasks, such as the peak analysis, differentially expressed gene identification, etc. Also, users caring about the figures that can be generated from the tool. Otherwise, users will still need to combine the use of Flexynesis with other packages.

6. Providing codebase, examples, and tutorial are normal standards for tool development nowadays. It is not a challenge. Flexynesis does not show innovative aspects to overcome this "challenge".

7. The target user of Flexynesis is unclear. For developers, for most of time, they do not care whether they are using a single all-in-one package or build their own pipeline with multiple packages. For biologists, a web-based code-free server definitely will be more user-friendly and better for reproducibility.

8. Why use single-cell data for a bulk multi-omics tool?

9. How are those markers identified via integrated gradients differ than differentially expressed genes (e.g., between responder and non-responders)?

10. The authors present several performance metrics for model evaluations but do not fully explain the implications of these results. For instance, what are the potential limitations of these performance indicators in clinical settings? How do these findings translate to real-world applicability?

11. The paper discusses feature selection using Laplacian Score but could benefit from a discussion on why this method was chosen over others. Are there plans to incorporate alternative feature selection techniques in future versions?

12. Some statements or terms used in the manuscript are overlaid. It would be more accurate to call it a pipeline instead of framework; it can be used for tumor samples analysis but definitely not deep enough for precision oncology and beyond.

(Remarks on code availability)

Version 1:

Reviewer comments:

Reviewer #2

(Remarks to the Author)

The authors made a significant effort and addressed the majority of the comments made so far. To this end, some benchmarking statistics would be of added value to the readership. Even though many models are compared, significance testing of performance differences (e.g., paired permutation or bootstrap tests) is not reported. Readers need to know if observed gains over RF/SVM are statistically meaningful.

Furthermore, please provide the exact priors/ranges used for each model class and a run-time summary (GPU/CPU hours). Search space boundaries and optimisation budget are only sketched at the moment (Bayesian sequential optimisation)

Of note, most demonstrations use public cell-line resources; only one real patient cohort (LGG/GBM survival) is shown. A truly clinical prediction task (e.g., therapy response in TCGA with held-out hospital data) would strengthen the "precision oncology" claim.

A related patent (US 18 274 271) is listed, yet the "Declaration of conflict of interest" says "None declared". Please clarify whether the patent covers Flexynesis and whether it limits permissive open-source use.

Runtime scalability table: please include wall-clock times for a standard dataset on CPU vs GPU.

Memory footprint: please give approximate RAM usage for early- vs intermediate-fusion.

(Remarks on code availability)

Reviewer #3

(Remarks to the Author)

I thank the authors' response and revision. I have no more questions.

Review comments on previous responses to Reviewer 1:

- Major Comment 1: The response to this comment is fine, but I think having the single cell example as the first example is not appropriate. As the authors admitted that applying Flexynesis to single cell data analysis is different than bulk data, it is more like an extension to the Flexynesis ability. It is confusing when nothing about single cell data was mentioned in the title or introduction and suddenly read a single cell example as the first result section. I suggest the authors replace the single cell example with a bulk data example and can move the single cell result to the later section.
- Major Comment 5: I suggest adding the hardware usage in the methods section and an additional Supplementary Figure.
- Major Comment 7: The Figure and comparison result should be included in the context.
- Source Data should be provided as supplementary excel files for all evaluations, comparisons, and some results (e.g., Figure 4C, Figure 6, Figure 7A-B, and Figure 8B).
- The color representations should be indicated in Figure 7B, not only in legend.

(Remarks on code availability)

Version 2:

Reviewer comments:

Reviewer #2

(Remarks to the Author)

The authors did their very best and addressed all comments made.

(Remarks on code availability)

Reviewer #3

(Remarks to the Author)

I have no further comments to the authors.

(Remarks on code availability)

Response to reviewers

We are grateful to all reviewers for their time and effort and wish to thank them for their constructive comments on the manuscript, which helped us make significant improvements. We have revised the manuscript accordingly and hope that the changes and our point-by-point response below are adequate. Our responses to comments are in **black** and reviewer comments are in **red**.

Reviewer #1 (Remarks to the Author): Expert in machine learning, deep learning, computational cancer genomics, multiomics, and drug response prediction

In this study, Bora et al. introduced Flexynesis, a comprehensive solution designed to enhance the utility and applicability of deep learning in multi-omics data analysis. Flexynesis streamlines data processing, enforces structured data splitting, and ensures rigorous model evaluation. This toolset could make deep-learning based bulk multi-omics data integration in the context of clinical/pre-clinical data analysis and marker discovery more accessible to a wider audience with or without experience in deep-learning development. The authors demonstrated the versatility of Flexynesis through various use cases, including drug response prediction, cancer subtype modeling, survival analysis, and biomarker discovery.

Indeed, methods for bulk multi-omics data integration are important, and this toolset could make deep learning-based multi-omics data analysis more accessible to wider users. So, this study is timely and very useful for the community. However, there are several technical limitations. Some issues are itemized below.

We thank the reviewer for appreciating our work and we are grateful for the constructive comments they provided, which we believe improved our manuscript and the software significantly. Based on the reviewer's comments, we have done the following improvements:

- We have added new features to Flexynesis:
 - Added XGBoost as an alternative model and re-ran our experiments including XGBoost where applicable.
 - We also provide AUROC and AUPR metrics as alternatives for measuring the prediction performance of our models for classification tasks.
 - We have added utility functions for batch integration of sample embeddings post-training using two different approaches: 1) reciprocal PCA with mutual nearest neighbors, 2) optimal transport
 - We enable the user to utilize clinical variables as covariates in the modeling procedure, where the clinical variables are processed and used as an additional input data modality.
 - We developed an alternative feature attribution method using GradientSHAP for all available neural network models.

- We added GNNs in additional use-cases (finetuning and drug response marker discovery sections).
- We did an ablation experiment for the hubness score prediction (with/without hubness) in cross-modality training to predict cell line gene dependency scores.
- We drew new diagrams for data processing steps, hyperparameter optimization, model finetuning, and different model architectures. The diagrams are available in the supplementary figures file.
- We have improved the figures for better presentation.

Major points:

1. The article's title mentions bulk multi-omics data integration, but subsequent case studies utilize single-cell data. Can the current pipeline be extended to single-cell multi-omics data integration? What are the limitations of the current pipeline in integrating single-cell multi-omics data?

We thank the reviewer for pointing this out and apologize for the confusion. Currently, we don't see Flexynesis as an alternative to single-cell-oriented tools. The reasons are fourfold. First reason is about the contrast between the main emphasis on supervised or unsupervised tasks for multi-omics integration. While single-cell methods are mainly driven by unsupervised approaches, Flexynesis is built mainly for supervised tasks. The second reason is about the assumptions of the technology that is generating the data. While single-cell methods are suitable for dedicated single-cell data generation technologies, we tried to build Flexynesis in a data-agnostic manner, where the data can be any kind of tabular data. Thirdly, single-cell methods are tuned to not just the types of data, but also the scale of data that can reach millions of cells, which requires dedicated data structures, while Flexynesis is not built for such scale, as our main goal was (pre-)clinical cohorts of bulk sequencing datasets which are on the order of thousands to tens of thousands of samples in the extreme case. Finally, the arguments we make about the features such as usability/packaging/documentation are usually not valid for many single-cell tools, as there are many such tools that have followed best practices in that regard.

A major difference in the kinds of applications where we utilize Flexynesis in contrast to the typical single-cell pipelines, is that there is a big emphasis on supervised tasks in (pre-)clinical cohort studies as there are always some clinical sample labels available that can guide the analysis, such as survival outcomes, disease subtypes, histology, patient characteristics such as age, gender, so on and so forth. On the other hand, the single-cell omics applications are usually driven by "unsupervised" applications, where individual cell identities are not always apparent, therefore unsupervised approaches for data integration and clustering is typically carried out, which is followed by differential marker analysis to ascribe identities to such clusters. Although we have also provided such functionalities for Flexynesis (unsupervised clustering using variational autoencoders, clustering, and associated utility functions for visualisation), *purely* unsupervised approaches are not our main goal, rather a side-product of what we would like to achieve with Flexynesis. Other single-cell oriented tools are better equipped for investigation of such unsupervised clusters with methods for unsupervised integration, marker analysis, and a variety of accessory utilities one needs to inspect such clusters of single-cells.

Even for applications of (pre-)clinical bulk sequencing cohorts where unsupervised analysis is carried out such as disease subtyping, we expect Flexynesis to be used in a *supervised* manner. For instance, to investigate prognostic disease subtypes, one can use Flexynesis with a semi-supervised approach as the model can be trained with supervisor MLPs that predict survival outcomes or MLPs that predict treatment outcomes. Thus the sample embeddings would reflect clusters that are guided by prognostic labels. Similar ideas can be applied for any subtyping scheme – whether molecular or prognostic. Thanks to the multitasking support, one can do clustering of samples across multiple patient covariates, thus delineating intersectional molecular/prognostic subtypes. So, even for the typically unsupervised tasks such as disease subtyping, we expect Flexynesis to be used in conjunction with some supervision associated with clinical variables.

Although we use Flexynesis mainly for multi-omics data integration, current implementation is built in a data-agnostic manner, where the only assumption is that the input data matrices are in a tabular format, in other words, it is a multi-modal data integration tool suite. Along this line of thought, we have utilized not just bulk omics data, but other kinds of tabular data such as protein-language model embeddings. With the same motivation, we also wanted to observe if it would work on supervised tasks on single-cell multi-omics, and it turned out to be useful for the cell type classification task using CITE-Seq data.

In summary, we don't see Flexynesis as an alternative to single-cell multi-omics data integration methods, but it could be used in conjunction with the available multi-omics integration tools particularly for supervised modeling tasks, for instance a dataset preprocessed using single-cell methods could be used as input to Flexynesis if there is a supervised task to use it on.

We updated the Discussion section to clarify this point.

2. Compared to single-task modeling, does multi-task modeling, while capable of predicting multiple outcome variables simultaneously, result in decreased performance for one specific outcome variable?

Yes, it is true that depending on which sets of outcome variables are used as targets, prediction performances for individual targets may either deteriorate or benefit from this. For instance, the usage of age and histological subtypes as additional target variables in prediction of survival outcomes for the glioma cohort increases the prediction accuracy. The reason is that such variables are complementary in the disease etiology and survival outcomes. However, even though we have implemented loss weighting strategies (see methods) when multiple loss values are involved, it could be the case that unrelated variables provided as targets could hurt each other's prediction performance. This actually ties to our main motivation to build Flexynesis, that the user can easily build multiple models in quick succession, test various hypotheses by merely changing the list of variables in the command-line argument (`--target_variables`). The user doesn't have to fiddle with the model structure, number of units, outcome variable types, which are all taken care of automatically. In short, whether multi-tasking provides benefit or not depends on the

selection of variables and how such variables are associated to each other in the specific context, which the user can easily test by merely listing different sets of variables.

3. Could you create a simple diagram of the omics data preprocessing steps to facilitate reader understanding?

We thank the reviewer for this suggestion. We have built diagrams for the data processing steps and provided them in the Supplementary Figures file (Supp. Figure 1).

4. How can the impact of batch effects be assessed and mitigated when integrating data from different batches or sources?

We thank the reviewer for this question. We have previously experimented with various approaches in addressing batch integration during training, such as adversarial training, gradient reversal approaches for batch variables, and applying various loss functions for the predictability of batch variables. However, in our prototypes, we could not land on a good solution that generalized well across use-cases for the bulk datasets in a model-agnostic and data-agnostic manner. To address this problem in a model-agnostic manner, we have added new features to the package for incorporating known batch variables in the prediction models. Also, we provide two different post-training utilities for batch-alignment of sample embeddings.

Firstly, we have developed a new feature to allow users to provide a list of covariates, which can be both batch-related variables and relevant clinical variables that could be informative in the prediction model. The input covariates are parsed from the clinical data file, processed (for instance categorical variables are one-hot-encoded) and used as an additional data modality to obtain sample embeddings. This strategy is motivated by the previous approaches: for instance DESeq2 does not use batch-corrected gene expression values for building the models for differential expression, rather it uses variables as covariates in the model. Also, a benchmarking study for biomarker discovery in single-cell methods has also reported that biomarker discovery does not benefit from batch correction, and they recommend using batch variables as covariates in the models (Nguyen H.C.T et al, Nat. Comms, 2023, <https://doi.org/10.1038/s41467-023-37126-3>). However, we also acknowledge that it is important to be able to align the sample embeddings to enable mapping of samples with common biological similarities from different datasets. So, we have built two utility methods, which can be used in a method-agnostic manner to align sample embeddings from different batches post-training. First method we provide employs reciprocal PCA analysis to align the batch embeddings, and the second one employs optimal transport to align embeddings from different batches.

Below is a demonstration of batch-alignment of embeddings post-training on a use-case, where we predict the cancer type, in a dataset comprised of a mixture of human tumor samples from three different cancer types (breast cancer, colorectal cancer, and glioblastoma) and CCLE cell lines derived from the same cancer types. The first row in the figure below shows the sample embeddings post-training before applying any batch alignment functions. After the training, we can observe that the samples can be distinguished by cancer type regardless of the data source (panel B), however there is a batch effect where the samples also cluster by the data source (TCGA vs CCLE - panel A).

Using reciprocal PCA with MNN (panel C and D) or Optimal Transport (E and F), the embeddings can be aligned by data source while preserving the cancer type clusters.

In this use-case, the networks learn to predict the cancer types regardless of the data source, however, these approaches could be utilized for visualisation purposes and to be able to map similar samples from different data sources.

5. It will be helpful to provide an approximate description of the hardware used and the runtime.

Below is a demonstration of the time it takes for different models to do a single hyperparameter optimization step on a typical dataset with 500 samples and ~2000 genes used as input features.

CPU specs: Intel(R) Xeon(R) Platinum 8168 CPU @ 2.70GHz

GPU specs: Nvidia Tesla P40

6. Given the excellent performance of XGBoost on tabular data, it is recommended to include it as a baseline.

We thank the reviewer for this suggestion. We have integrated XGBoost as an additional method option and updated our analyses where we included XGBoost in our comparisons where applicable.

7. In the “Marker discovery” section, the title does not reflect the evaluation of feature importance. it is recommended to clearly specify the model's functions and applications in the title.

Moreover, feature importance evaluation is often unstable; could additional methods besides IG, such as SHAP, be included to assess whether the results are consistent with IG?

We thank the reviewer for both suggestions. We have modified the title to reflect the content: “Discovering biomarkers of drug response in cell lines”. Moreover, we have added an alternative means of computing feature attribution scores using a GradientSHAP method (<https://github.com/BIMSBbioinfo/flexynesis/pull/98>). The users can now use the “--feature_importance_method” flag to choose between “IntegratedGradients” and “GradientSHAP”. We observed that for a given trained model, the two methods have a very high level of agreement for feature importance scores (not identical but close to perfect agreement for this case). Such a high agreement between the two methods is presumably due to both methods taking advantage of the model gradients to compute the feature attributions. Below is a scatterplot of attribution scores for a DirectPred model trained on Erlotinib drug response prediction markers:

The reason we implemented GradientSHAP rather than the classical SHAP method is that while the SHAP method is deterministic as it is applied on all possible subsets of features, it becomes computationally infeasible to use when there are thousands of features available. Hence, methods that approximate SHAP have been developed to suit applications in deep neural networks. So, we chose this method (https://captum.ai/api/gradient_shap.html), which was actually implemented by the primary author of SHAP.

8. The test cases do not include GNN models; however, GNN methods have shown certain advantages in previous studies. It is recommended to incorporate GNN models in some case studies.

We have included GNNs in further use cases for the demonstration of fine-tuning and the demonstration of drug response marker discovery sections and updated the figures and main text accordingly.

9. In “Cross-modality learning” section, “In addition, we attached a supervisor MLP to guide the network to predict the hubness-score of each gene in the genetic interaction networks obtained from the STRING database assuming that the centrality of a gene in biological interaction networks could be a contributing factor in its essentiality for cell survival.”

An ablation experiment is needed to demonstrate that the inclusion of the supervisor MLP indeed aids in reconstructing the cancer cell line dependency scores.

We have re-run the models both with and without the hubness score as a supervisor variable. We observe that including this as an additional variable improves the cross-modality reconstruction performance in the single-modality case, however leads to a deterioration when all three modalities were used at the same time, which could be because the network may have put more weight on learning the “hubness” feature while the weights on cross-modality reconstruction may have been diluted with the addition of further data modalities in this particular case.

We have updated Figure 5 to include a diagram of how this training works (See also Supplementary Figure 6 for a more detailed diagram of cross-modality prediction networks). We also simplified the figure to only report the performance on the test samples for better presentation purposes.

10. “During model training, the training data is split by default into 80/20 portions for training and validation. The user can also select to do a k-fold cross-validation, in which the training data will be split into k-folds.”

Could an independent test set be split for evaluating model performance, such as an 80/10/10 split? and then use k-fold cross-validation for hyperparameter optimization, use independent test set for evaluation.

As the data preparation is the responsibility of the user, the user can split their input data as they wish (e.g. use 10% for evaluation/test set) and use 90% as input for training. This 90% portion is then split as train/validation splits by flexynesis (by default 20% of 90% would be used for validation, however, the user can change the size of validation proportion to be used as they wish with the “--val_size” argument) or can be used in a k-fold cross-validation scheme.

Minor points:

1. For some of the more complex model setups, such as cross-modality learning, could you provide a simple model architecture diagram to facilitate reader comprehension?

We have drawn further diagrams for all the steps from data import, different model architectures, and parameter tuning schemes (hyperparameter optimisation and model fine-tuning) and provided them in the supplements (Supplementary Figures 1-8).

2. Why were AUC and AUPR not used in the evaluations of classification tasks?

We thank the reviewer for the suggestion. We have included additional model performance evaluation metrics for the classification tasks including “AUROC” and “AUPR”. For categorical variables with more than 2 classes, these values are computed in a one-vs-rest fashion and weighted by the size of the corresponding categories, thus weighted averages of these scores are provided as model performance metrics.

3. In “Improving model performance via model fine-tuning” section, what is the specific architecture of the supervised-VAE model?

“We observed that, while fine-tuning does not guarantee an improvement in model performance it can provide a boost in accuracy (Figure 6A).” This sentence is confusing.

We have drawn diagrams to clarify the steps in how model fine-tuning works for any model (Supplementary Figure 3) and the architecture of the supervised-VAE model (Supplementary Figure 5). We have re-run this experiment including GNNs and also other baseline methods and updated the figure. What we mean by this sentence is that fine-tuning can help improve model performance in certain cases, but the impact doesn’t look significant for the use case where we trained the model on CCLE dataset and fine-tuned on different size portions of the GDSC dataset. The reason fine-tuning doesn’t show significant changes for this use-case can be explained by the fact that CCLE and GDSC datasets have similar distributions. To improve the presentation of this figure, we only kept one panel in the main figure (Erlotinib) and moved the remaining panels for the other drugs to Supplementary Figure 9.

In the second example use case, we had previously demonstrated a model trained on human tumor samples for neuroblastoma and fine-tuned it on neuroblastoma cell lines. Although this was not requested by the reviewer, we have taken the liberty to update this use-case demonstration to a similar setup with more samples where we could demonstrate this in a multi-omics setting. In the previous iteration (the neuroblastoma use case), we had only utilized “gene expression” data and we had few cell lines as the target dataset (~30 samples). We realized the results of this modeling can be sensitive to the chosen subset of samples for fine-tuning, so we decided to use a larger dataset with a multi-omic setting.

Therefore, we have replaced this analysis with a similar setup, where we used human tumor samples from the TCGA dataset for three different cancer patient cohorts (breast cancer, colorectal cancer, and glioblastoma) and chose CCLE cell lines from corresponding disease types. We built models to predict cancer types of TCGA samples and evaluated the models on predicting the cancer types of the cell lines. Without fine tuning, the models (both deep learning and classical approaches) fail to predict the cancer types of the cell lines. However, using a portion of the cell lines for fine-tuning, the models learn to predict the cancer types of the remaining test samples (See Figure 6B).

4. In figure 3A, using different background colors for high-risk and low-risk groups would be more effective for presentation.

We have implemented this change in Figure 3A.

5. In figure 4C, what does the colors in the bar represent?

We have updated the figure to include a legend title: “Concordance Percentage”. The color bar represents the percentage of concordance between detected clusters and the corresponding cancer types. While most clusters correspond to a single cancer type dominantly (high concordance: “red”), some cancer types are split into multiple subtypes, which leads to lower (blue) concordance percentages.

6. There is significant room for improvement in the visualization of the figures.

We have followed the specific suggestions of the reviewer and also applied some reorganization to improve the presentation of the figures where possible. Moreover, we have provided many additional diagrams to help improve the graphical representation of the processes involved in obtaining the figure results as supplementary figures.

Reviewer #3 (Remarks to the Author): Expert in single-cell multiomics, bioinformatics, machine learning, deep learning, and drug development

The manuscript presents Flexynesis, a deep learning pipeline aimed at integrating bulk multi-omics data, enhancing usability and adaptability for various precision oncology applications. The authors emphasize the limitations of existing methods, such as lack of transparency, modularity, and narrow applicability, which Flexynesis seeks to address. They demonstrate the framework's capabilities across various modeling tasks including regression, classification, survival analysis, and biomarker discovery, along with a benchmarking pipeline for model performance evaluation. Overall, the novelty of the underlying algorithms is very limited, with an emphasis on existing tool integration and optimization rather than development of new methods. It lacks sufficient explanations for method selection and more.

1. The Introduction section needs to be revised to make it sharp and concise. The logic and core message are not clear.

While we think the text already reflects the major motivations, the limitations of existing tools in the similar research space, and the proposed features to solve such problems, we thought the addition of more graphical diagrams could improve understanding of the text. Therefore, we have built various diagrams starting from how the initial data is imported, processed, how it is trained in various different selected architectures, how hyperparameter optimization works and how we do fine-tuning of trained models (See Supplementary Figures 1-8). We refer to these diagrams within the text where applicable. We hope these diagrams will improve the understanding of the manuscript.

2-3. While the manuscript outlines the importance of multi-omics data integration in precision oncology, it could strengthen its case by providing more detailed examples of how Flexynesis has improved real-world clinical outcomes or research findings. What specific applications have demonstrated significant advancements due to this tool?

The true innovation of this work is unclear. No new methods, scores, or standards is proposed and show superior performance than existing ones.

We thank the reviewer for raising these questions.

Our main issue with the existing literature was not with the reported performance of existing models, rather with the inability to quickly reproduce the reported results, ascertain their robustness, and compare them to other methodologies. We have therefore constructed a standardized framework, where one can quickly form and test various hypotheses in the context of disease cohort analyses using both deep learning and classical machine learning methods. Besides the reconstructed use-cases, a novel use-case we have highlighted is the application of cross-modality networks, where we combine omics data with protein language embeddings to improve prediction performance of cell line gene dependency scores. To the best of our knowledge, this hasn't been achieved before.

It is true that none of the components we use in Flexynesis are novel. As a matter of fact, our final remark in the manuscript is this:

“As a final remark, it is important to note that what we developed here is not a set of novel deep learning algorithms. None of the components we built are novel, however the innovation comes from how these components are brought together into a usable package. Flexynesis improves user experience and makes multi-omic deep learning accessible to a broader audience”.

We have strived to put together methods that have been established in the literature for various use cases. We based our selection of tools based on previously published benchmarking methods (cited in the introduction) and the collection of many different published methods (as we listed in the Supplementary Table 1). We'd like to highlight the spirit of this argument with a quote from the book “Memories of a Nation” by Neil McGregor where he talks about the impact of the printing press in the 15th century and the context which Gutenberg had invented this machine:

“Each element in the complex process of producing a book existed already. Gutenberg's genius lay in combining all these various different processes into one coherent work”.

Of course, we don't claim that Flexynesis is as revolutionary as the printing press, we believe the underlying principle is along the same spirit, where the application of modern machine learning to multi-modal data analysis is made more accessible to a wider audience, rather than the invention of better components. With this toolkit, we can quickly frame many different hypotheses and combinatorially address them in a standardized workflow. We will use this framework for building more flavors of deep learning applications and the existing methods will serve as a baseline to build upon.

4. The authors state that Flexynesis supports various architectures, yet there is a lack of comparative analysis on why certain architectures were chosen over others for specific tasks. Also, is Flexynesis flexible to include any new models developed in the future? How?

We thank the reviewer for the question. If the reviewer is asking about the choice of architectures for the specific use-cases demonstrated in the paper, we actually run multiple architectures on each use-case, but pick the best performing model. However, in these

use-cases, all models performed relatively well, so the emphasis was not particularly on the model selection, but rather on how these models could be used in specific use cases (as the models can be easily interchanged by the user). In case the reviewer is asking about why these models were implemented as part of the package in general, the reason is that we tried to develop as many different flavors of deep learning models as possible, but we strived to pick models that have already been established in the literature. These established models will serve as a baseline to improve upon for the future developments of novel architectures.

We tried to build the tool in a modular structure, so that different flavors of models can be integrated with the package in a straightforward manner (assuming expertise in deep learning model development). Each model architecture is a pytorch-lightning class, which is self-contained, written as a single python script (see <https://github.com/BIMSBbioinfo/flexynesis/tree/main/flexynesis/models>). All classes take the same standardized multi-modal dataset as input. Each model must have a hyperparameter optimisation configuration, a list of target variables to automatically build required MLPs for the respective supervision tasks, and methods to make predictions on a given new dataset, transform a collection of samples to the model's embedding space, and also compute feature attribution scores (marker finding module). With these constraints in place, any developer can add new architectures or variants of existing architectures to our toolkit.

5. It is not clear the output of Flexynesis. One of the gaps Flexynesis wants to fill is the limited tasks. However, Flexynesis also only focuses on five tasks of regression, classification, survival, clustering, and cross-modality prediction. If the author considers Flexynesis as an all-in-one or one-stop tool for bulk multi-omics, then, more analyses should be included, especially those omics-specific tasks, such as the peak analysis, differentially expressed gene identification, etc. Also, users caring about the figures that can be generated from the tool. Otherwise, users will still need to combine the use of Flexynesis with other packages.

With “task”, we are referring to the typical list of machine learning problems, such as classification, regression, time-to-event (survival) modeling, clustering, and so on and so forth. So, Flexynesis covers all these typical tasks in one framework, where the user doesn't have to worry about building their own loss functions or specific data transformations required for each outcome variable and type of task. None of the other multi-omics integration methods (also listed in Supplementary Table 1) can cover all of these in one framework.

A “peak analysis” would be a preprocessing analysis step for a multi-omics project. Peaks identified for a cohort of samples could be used as input to Flexynesis. For the differentially expressed gene identification, we provide dedicated modules to rank features based on which features contribute most to the learning process.

We clarified the boundaries of Flexynesis by adding diagrams starting from data import up to fine-tuning of models in supplementary figures.

Regarding visualisation capabilities and other accessory utilities, we have developed various utilities for analysis and visualisation of sample embeddings using dimensionality reduction methods (PCA/UMAP), scatter plots, box plots, Kaplan-Meier curves, hazard ratios for survival analysis, and heatmaps for concordance between predictions and known values. We have a variety of associated utilities which can be seen in <https://github.com/BIMSBbioinfo/flexynesis/blob/main/flexynesis/utils.py>. Also, we have demonstrated the usage of such visualization methods in various use cases in tutorials in the form of jupyter notebooks, which can be found at: <https://github.com/BIMSBbioinfo/flexynesis/tree/main/examples/tutorials>

6. Providing codebase, examples, and tutorial are normal standards for tool development nowadays. It is not a challenge. Flexynesis does not show innovative aspects to overcome this “challenge”.

We respectfully disagree with the reviewer on this aspect. Unfortunately, the published literature on this space is teeming with methods that don't come with re-usable tools, codebase, tutorials, and/or documentation (we have listed about 80 different papers in the Supplementary Table 1). However, it could be true that in some sub-fields, such practices are better followed, for instance, many single-cell multi-omics integration methods are properly maintained and documented, so entry into such frameworks may not be as difficult for the average bioinformatician.

7. The target user of Flexynesis is unclear. For developers, for most of time, they do not care whether they are using a single all-in-one package or build their own pipeline with multiple packages. For biologists, a web-based code-free server definitely will be more user-friendly and better for reproducibility.

We strived to make it usable for anyone that has some programming experience.

We have built a graphical user interface by integrating Flexynesis with the main Galaxy server, which is maintained in Freiburg, Germany. We developed a tutorial on how to use it on the Galaxy server:

https://bimsbstatic.mdc-berlin.de/akalin/buyar/flexynesis/site/running_on_galaxy/

We also strive to make it easy for the computational scientists by making it a tool that can be easily integrated with pipelines. We understand the sentiment that a developer could just build their own custom deep learning models, however, this is an endeavor that takes a lot of time even for experienced developers, which is why we developed this suite for our own research projects, because it streamlines quickly addressing questions such as “which data modalities are best for this task”, “which variables are most informative”, “which deep learning approach is overall better”, “do we really need deep learning for this task, could we just as well use an SVM?”. In essence, this tool could be used by any computational scientist, where they don't have to excel in deep learning development.

8. Why use single-cell data for a bulk multi-omics tool?

We thank the reviewer for raising this question. We have received the same question from Reviewer #1, so we provide a shorter version of the same answer here. In essence, we

wanted to test it on a classification problem, where the CITE-Seq data looked like an interesting test case for predicting cell type labels.

Currently, we don't see Flexynesis as an alternative to single-cell-oriented tools. The reasons are fourfold. First reason is about the contrast between the main emphasis on supervised or unsupervised tasks for multi-omics integration. While single-cell methods are mainly driven by unsupervised approaches, Flexynesis is built mainly for supervised tasks. The second reason is about the assumptions of the technology that is generating the data. While single-cell methods are suitable for dedicated single-cell data generation technologies, we tried to build Flexynesis in a data-agnostic manner, where the data can be any kind of tabular data. Thirdly, single-cell methods are tuned to not just the types of data, but also the scale of data that can reach millions of cells, which requires dedicated data structures, while Flexynesis is not built for such scale, as our main goal was (pre-)clinical cohorts of bulk sequencing datasets which are on the order of thousands to tens of thousands of samples in the extreme case. Finally, the arguments we make about the features such as usability/packaging/documentation are usually not valid for various single-cell tools, as there are a variety of such tools that have followed best practices in that regard.

We updated the Discussion section to clarify this point.

9. How are those markers identified via integrated gradients differ than differentially expressed genes (e.g., between responder and non-responders)?

When looking for markers that distinguish responders from non-responders, one could carry out a differential expression analysis for transcriptomics data. However, in the context of multi-modal analysis where data modalities exist other than transcriptomics, such approaches are not applicable. When making predictions from multiple omics modalities, interactions between omics variables can provide additional information with regards to the given task. For example, a gene which was highly differentially expressed between healthy and diseased samples, might be less informative for classification when compared with CNV + mutation + expression of three different genes.

When a neural network is trained to predict an outcome, an established approach for marker discovery (alias feature attribution scoring in the context of neural networks) is to use various explainability methods to find which features contributed most to the outcome prediction performance. Such an approach to ranking markers by importance is data-agnostic and is applicable for deep learning models where an integration of multi-modal data is required. Integrated Gradients is one among a variety of such methods. We chose this because it is an established method efficient enough to handle the typical datasets that are dealt with (thousands to tens of thousands of samples with thousands to tens of thousands of features). With the request from Reviewer #1, we have also implemented GradientSHAP method as an alternative for feature attribution scoring.

10. The authors present several performance metrics for model evaluations but do not fully explain the implications of these results. For instance, what are the potential limitations of these performance indicators in clinical settings? How do these findings translate to real-world applicability?

The performance metrics we use are established metrics that are used in evaluating the performance of machine learning models. We use various metrics because some of them can be biased for data imbalance or data spread. Using such metrics helps prioritizing if a model performed well enough to trust the biomarkers derived from these models. For real-world applicability, we evaluate the models on test datasets that are unseen during training, which helps prioritizing the most successful models, hence markers derived from those models should have higher credibility. With the drug response marker discovery use-case, we demonstrate that the known targets of the drugs are re-discovered (however retrospectively), which just shows that the model could be used for such applications where the markers are not known.

This manuscript is more about demonstration of the tool in finding expected results across various use-cases. Best we can do with a software model is to make sure it is as unbiased as possible and has the utilities to help prioritize the best models and reliable markers. The reason we also utilized various baseline models out of the box (XGBoost, Random Forests, SVMs) is to easily test whether deep learning provides any added value in the specific dataset and task. However, for real clinical applications, the biomarkers derived from these models would need to be further investigated in the clinical setting.

11. The paper discusses feature selection using Laplacian Score but could benefit from a discussion on why this method was chosen over others. Are there plans to incorporate alternative feature selection techniques in future versions?

The reason we have chosen Laplacian Score is because it is an unsupervised feature selection method, where the feature volume can be decreased without changing the features into abstract forms. Machine learning methods' performance can often decrease when many input features are highly correlated. Laplacian scoring directly mitigates this issue. Additionally, the method doesn't lead to latent features (as is done with dimension reduction techniques such as PCA), thus the original features are preserved. We needed an unsupervised approach here, because the user could be interested in various different combinations of target variables or apply it on different model architectures. To provide a common standardized input to the downstream steps, we chose this approach, which is fast and efficient for the dataset scale of the use-cases we have demonstrated.

There are alternative methods which would be interesting to incorporate such as concrete autoencoder or fractal autoencoders, which we mentioned in the discussion section.

12. Some statements or terms used in the manuscript are overladen. It would be more accurate to call it a pipeline instead of framework; it can be used for tumor samples analysis but definitely not deep enough for precision oncology and beyond.

The choice of "framework" might not be completely appropriate. Maybe a better naming could be "toolkit" or "suite of tools". On the other hand, we don't think "pipeline" is not a very good choice, either, as it is a standalone tool and can be used as a building block for a larger pipeline, which is what we did to build the flexynesis-benchmarking pipeline.

The choice of the words ".. and beyond" was to reflect the fact that this toolkit can be used on any multi-modal data analysis context as long as the inputs are tabular and the samples

are represented across the modalities. So, it doesn't have to be only applied for cancer research or tumor samples. All the tasks such as survival modeling, classification, regression, clustering can be relevant in any disease context. The primary reason we used "precision oncology" in the title is because the use-cases were selected for this context.

Reviewer #2 (Remarks to the Author): Expert in biomarkers, pharmacogenomics, and precision medicine

The authors made a significant effort and addressed the majority of the comments made so far.

We thank the reviewer for constructive comments and we are glad that we could resolve the previous major concerns.

To this end, some benchmarking statistics would be of added value to the readership. Even though many models are compared, significance testing of performance differences (e.g., paired permutation or bootstrap tests) is not reported. Readers need to know if observed gains over RF/SVM are statistically meaningful.

We implemented paired bootstrap tests to enable comparisons between methods. For each pair of methods in comparison, we generate 100x resamplings (with replacement) of holdout samples, evaluate the models, compare the performances using paired t-tests.

We have made updates in the fine-tuning analysis (Figure 6) and general benchmarking results (Figure 8), where we made direct comparisons of deep learning models with classical machine learning models. We also provide the bootstrap testing results in Supplementary Table 4 (for fine tuning analysis) and Supplementary Table 6 (for the general benchmarking results).

For Figure 6, we have compared the best performing deep learning model with fine tuning against non-fine-tuned deep learning models and classical machine learning models. For the drug response models (Figure 6A shows one example drug analysis: Selumetinib), the best deep learning model doesn't show a clear improvement over non-fine-tuned deep learning models or classical machine learning models. Although we can get low p-values from paired t-tests, which picks up even slight improvements in model performance, the effect sizes are rather small. The previous conclusions we have made still hold for this analysis, which is that fine tuning doesn't bring extra value in this pair of datasets (CCLE vs GDSC) where the data distributions are rather similar. On the other hand, the same analysis for prediction of cancer types trained on human tumor samples and predicted on cell lines (CCLE) does not work without fine-tuning in either deep learning models or classical machine learning models (Figure 6B). The fine-tuned deep learning model is clearly better. We have deposited the paired bootstrap statistics in Supplementary Table 4.

For Figure 8, we re-ran the benchmarks (this time focusing on only multi-omics settings, excluding single-omics experiments). For each of the 14 different modeling tasks, we carried out a paired bootstrap test between the best performing deep learning model and the best performing baseline model (Random Forest, SVM, XGBoost) and reported the bootstrap statistics in

Supplementary Table 6 (Sheet 2). From this analysis, we observe that deep learning models are better in 7 out of 14 tasks, while baseline methods are better in 6 out of 14 tasks and they have a tie in 1 out of 14 tasks. However, it would be safe to conclude that the performances are overall comparable in most tasks. It is still probably best to try multiple models from both groups and use the best performing one for the dataset at hand. The decision whether to use deep learning should probably not be solely dependent on the prediction performance metrics, but also other flexibilities offered by these methods such as transfer learning via fine tuning, tolerance for missing labels, multi-task modeling, dimensionality reduction capabilities, and different fusion strategies.

Furthermore, please provide the exact priors/ranges used for each model class and a run-time summary (GPU/CPU hours). Search space boundaries and optimisation budget are only sketched at the moment (Bayesian sequential optimisation)

In our experiments, we used the default hyperparameter search spaces implemented in the source code of Flexynesis (see <https://github.com/BIMSBbioinfo/flexynesis/blob/main/flexynesis/config.py>). We have added a table ("Table 1: Hyperparameter Optimization Search Spaces") in the methods section under the "Hyperparameter Optimisation" subsection that lists the parameter search space configurations for different model architectures.

Of note, most demonstrations use public cell-line resources; only one real patient cohort (LGG/GBM survival) is shown. A truly clinical prediction task (e.g., therapy response in TCGA with held-out hospital data) would strengthen the "precision oncology" claim.

While it is true that we have extensively utilized the pharmacogenomics datasets of cancer cell lines in our manuscript due to their importance in pre-clinical research in the context of precision oncology, their ease of public access, and their well standardized data formats, we have also extensively utilized publicly available human cancer cohorts in our manuscript.

Besides the survival and marker analysis of LGG/GBM cohorts (Figure 3), we have demonstrated a use-case of multi-task classification of metastatic breast cancer (METABRIC study) subtypes conditioned on treatment status (Figure 2); we have demonstrated unsupervised analysis of 20 human cancer cohorts from the TCGA dataset (Figure 4); we have demonstrated impact of model fine-tuning using human colorectal cancer, glioblastoma, and breast cancer samples (in combination with CCLE cell lines) (Figure 6B).

During the revision phase, we have added one more use-case utilizing human tumor samples from pan-gastrointestinal and gynecological cancers to classify samples with high microsatellite instability (MSI) (Figure 1B). What makes this use-case interesting is we could demonstrate that such samples can be detected using only gene expression data or in combination with methylation data (Figure 1B, Supplementary Table 2). Patients with high MSI status are more likely to

respond to immunotherapies, therefore this should qualify as a further example application in the context of precision oncology.

The reason we emphasize “precision oncology” in the manuscript is that we utilize Flexynesis on various use-cases that are typically employed in this field such as subtype analysis, survival prediction, and unsupervised clustering, where patients are stratified into prognostic/diagnostic groups using molecular data modalities. We also constrained our use-cases to cancer-specific multi-omics datasets. However, the tool itself is designed in an omics-agnostic manner, therefore it could be utilized in any other kind of disease where there are sufficient multimodal datasets (in data matrix format).

A related patent (US 18 274 271) is listed, yet the “Declaration of conflict of interest” says “None declared”. Please clarify whether the patent covers Flexynesis and whether it limits permissive open-source use.

We provide a conflict of interest statement in the manuscript section “declaration of conflict of interest”, where we mention the patent. A subset of the ideas implemented in Flexynesis are included in the patent application, however Flexynesis as a tool is not. The patent application predates the development phase of the software. The tool is freely available for academic usage. We only ask for permission for commercial usage. The tool and associated course work has been freely made available for the community (see https://github.com/BIMSBbioinfo/compngen_course_2025_module3).

Runtime scalability table: please include wall-clock times for a standard dataset on CPU vs GPU.

Memory footprint: please give approximate RAM usage for early- vs intermediate-fusion.

We re-did a resource profiling experiment, in which we used 500 breast cancer samples with gene expression and copy number alteration data modalities consisting of 2000 features each. We profiled the time (wall clock time) it takes to import data and run a single hyperparameter optimization step. We also profiled the CPU (system) RAM and GPU RAM statistics for 5 different neural network architectures with two different fusion approaches (early/intermediate), where applicable. We added Supplementary Table 7 and Supplementary Figure 10 to summarize these statistics. We observe that DirectPred (classical feed forward network) with intermediate fusion utilized with GPU is the fastest option. Models with intermediate fusion have smaller number of parameters, therefore faster to train and consume less memory. We added a sub-section titled “Runtimes and resources” under the “Methods” section explaining this procedure and results.

Reviewer #3 (Remarks to the Author):

I thank the authors' response and revision. I have no more questions.

We thank the reviewer for the constructive comments and we are glad to have clarified the reviewer's concerns over the previous version of our manuscript.

Review comments on previous responses to Reviewer 1:

• Major Comment 1: The response to this comment is fine, but I think having the single cell example as the first example is not appropriate. As the authors admitted that applying Flexynesis to single cell data analysis is different than bulk data, it is more like an extension to the Flexynesis ability. It is confusing when nothing about single cell data was mentioned in the title or introduction and suddenly read a single cell example as the first result section. I suggest the authors replace the single cell example with a bulk data example and can move the single cell result to the later section.

We replaced the single cell analysis use-case with a bulk data analysis use case in which we built a classifier for detecting samples with high microsatellite instability (MSI) using gene expression and promoter methylation data in seven TCGA datasets including gastrointestinal and gynecological cancer cohorts. We updated the relevant parts of the text and figure legends accordingly. We only kept the cell type classification task from CITE-Seq data as part of our benchmarking experiments (Figure 8).

• Major Comment 5: I suggest adding the hardware usage in the methods section and an additional Supplementary Figure.

As has been also requested by the other reviewer, we did a resource profiling experiment, in which we used 500 breast cancer samples with gene expression and copy number alteration data modalities consisting of 2000 features each. We profiled the time (wall clock time) it takes to import data and run a single hyperparameter optimization step. We also profiled the CPU (system) RAM and GPU RAM statistics for 5 different neural network architectures with two different fusion approaches (early/intermediate), where applicable. We added Supplementary Table 7 and Supplementary Figure 10 to summarize these statistics. We observe that DirectPred (classical feed forward network) with intermediate fusion utilized with GPU is the fastest option. Models with intermediate fusion have smaller number of parameters, therefore faster to train and consume less memory. We added a sub-section titled "Runtimes and resources" under the "Methods" section explaining this procedure and results.

• Major Comment 7: The Figure and comparison result should be included in the context.

We provide Supplementary Figure 9, in which we provide scatter plots comparing the feature importance attributions of the Integrated Gradients and GradientShap methods for each of the drugs analysed in Figure 7B and we refer to the supplementary figure in the relevant section of the main text.

- Source Data should be provided as supplementary excel files for all evaluations, comparisons, and some results (e.g., Figure 4C, Figure 6, Figure 7A-B, and Figure 8B).

We have created additional supplementary tables (Supp. Tables 2-6), where we printed the source data for the figures. For reproducibility purposes, we have also created a github repository containing all Flexynesis outputs, preprocessed datasets along with scripts for data preparation and scripts for reproducing the manuscript figures, which can be found at https://github.com/BIMSBbioinfo/flexynesis_manuscript.

- The color representations should be indicated in Figure 7B, not only in legend.

We updated the figure to add a legend in the figure to denote color representations for different omics layers.